# Explaining, Fast and Slow:
# Abstraction and Refinement of Provable Explanations

**Shahaf Bassan** [* 1]   **Yizhak Yisrael Elboher** [* 1]   **Tobias Ladner** [* 2]   **Matthias Althoff** [2]   **Guy Katz** [1]

## Abstract

Despite significant advancements in post-hoc explainability techniques for neural networks, many current methods rely on heuristics and do not provide formally provable guarantees over the explanations provided. Recent work has shown that it is possible to obtain explanations with formal guarantees by identifying subsets of input features that are sufficient to determine that predictions remain unchanged using neural network verification techniques. Despite the appeal of these explanations, their computation faces significant scalability challenges. In this work, we address this gap by proposing a novel abstraction-refinement technique for efficiently computing provably sufficient explanations of neural network predictions. Our method *abstracts* the original large neural network by constructing a substantially reduced network, where a sufficient explanation of the reduced network is also *provably sufficient* for the original network, hence significantly speeding up the verification process. If the explanation is insufficient on the reduced network, we iteratively *refine* the network size by gradually increasing it until convergence. Our experiments demonstrate that our approach enhances the efficiency of obtaining provably sufficient explanations for neural network predictions while additionally providing a fine-grained interpretation of the network's predictions across different abstraction levels.

## 1. Introduction

Despite the widespread use of deep neural networks, they remain uninterpretable black boxes to humans. Various

[*]Equal contribution   [1]Hebrew University of Jerusalem [2]Technical University of Munich. Correspondence to: <shahaf.bassan@mail.huji.ac.il>, <yizhak.elboher@mail.huji.ac.il>, <tobias.ladner@tum.de>.

*Proceedings of the 42$^{nd}$ International Conference on Machine Learning*, Vancouver, Canada. PMLR 267, 2025. Copyright 2025 by the author(s).

methods have been proposed to explain neural network predictions. Classic additive feature attributions such as LIME (Ribeiro et al., 2016), SHAP (Lundberg & Lee, 2017), and integrated gradients (Sundararajan et al., 2017) assume that neural networks exhibit near-linear behavior in a local region around the interpreted instance. Following these works, methods like Anchors (Ribeiro et al., 2018) and SIS (Carter et al., 2019) aim to compute a subset of input features that is (nearly) sufficient to determine the prediction. We refer to this subset of features as an *explanation* of the prediction. A common assumption in the literature is that a *smaller* explanation provides a better interpretation, thus, the minimality of the explanation is also a desired property (Ignatiev et al., 2019; Carter et al., 2019; Darwiche & Hirth, 2020; Ribeiro et al., 2018; Barceló et al., 2020).

Methods like Anchors and SIS rely on probabilistic sampling of the input space and lack formally provable guarantees for the sufficiency of the subsets they identify. In contrast, recent approaches have demonstrated that incorporating neural network verification tools can produce explanations that are provably certified as sufficient (Wu et al., 2023; Bassan & Katz, 2023; La Malfa et al., 2021; Izza et al., 2024). This makes such explanations more suitable for safety-critical domains where formally certifying the explanation is vital (Marques-Silva & Ignatiev, 2022).

Although such explanations are compelling, producing them is computationally expensive for large neural networks (Barceló et al., 2020) as they are obtained by solving neural network verification queries, which were shown to be NP-complete (Katz et al., 2017; Sälzer & Lange, 2021). While there have been rapid advances in the scalability of neural network verification techniques in recent years (Wang et al., 2021b; Brix et al., 2023), scalability remains a major challenge. Furthermore, providing *minimal* sufficient explanations requires invoking multiple verification queries: for example, methods suggested by previous research (Bassan & Katz, 2023; Wu et al., 2023) dispatch a linear number of queries relative to the input dimension, making these tasks particularly difficult for high-dimensional input spaces.

**Our Contributions.** In this work, we propose a novel algorithm that significantly enhances the efficiency of generating provably sufficient explanations for neural networks.

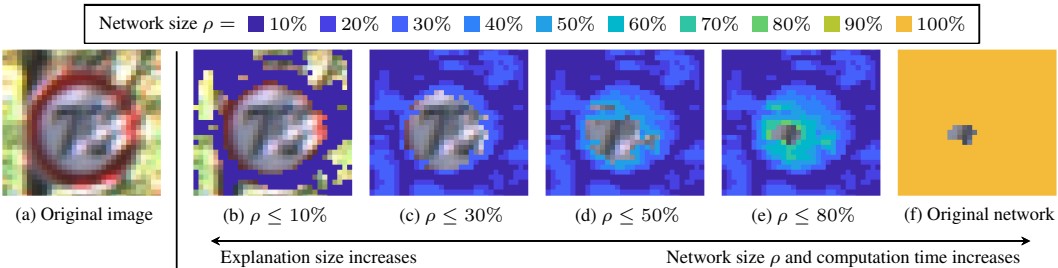

Figure 1: Demonstration of an abstraction-based explanation process. As the size of the abstract network $\rho$ increases, the size of the explanation (uncolored pixels) decreases. Notably, the majority of the explanation can be derived using only a small percentage of the network (b)-(e), reducing the time required to compute the explanation and offering more insight compared to using only the original network (f). Further visualizations are provided in Appendix C.

Our algorithm is based on an *abstraction refinement* technique (Clarke et al., 2000; Wang et al., 2006; Flanagan & Qadeer, 2002), which is widely used to improve the scalability of formal verification and model checking. In abstraction-refinement, a complex model with many states is efficiently optimized through two steps: (i) *abstraction*, which simplifies the model by grouping states, and (ii) *refinement*, which increases the precision of the abstraction to better approximate the original model.

In the context of explainability, we propose an algorithm that constructs an *abstract neural network* — a substantially reduced model compared to the original. This reduction is achieved by merging neurons within each layer that exhibit similar behavior. The key component of this approach is to design the reduction such that a sufficient explanation for the abstract network is also provably sufficient for the original network. Hence, we define an explanation for the abstract neural network as an *abstract explanation*. Verifying the sufficiency of explanations for an abstract neural network is much more efficient than for the original model due to its reduced dimensionality. However, if a subset is found to be insufficient for the abstract network, its sufficiency for the original model is undetermined. Consequently, while sufficient explanations for the abstract network will be sufficient for the original network, *minimal* sufficient explanations for the abstract network, though sufficient, may not be minimal for the original network.

To address this issue, we incorporate a refinement component as typical in abstraction-refinenemt techniques: We construct an intermediate abstract network, which is slightly larger than the initial abstract network but still significantly smaller than the original. The explanations computed for this refined network are still provably sufficient for the original network and are also guaranteed to be a subset of the explanation from the initial abstract network. Hence, this phase produces a *refined explanation* based on the refined network. After several refinement steps, the sizes of the neural networks will gradually increase while the sizes of the

explanations will gradually decrease until finally converging to a minimal explanation for some reduced network, which is also provably minimal for the original neural network. An illustration of this entire process is shown in Fig. 1.

We evaluate our algorithm on various benchmarks and show that it significantly outperforms existing verification-based methods by producing much smaller explanations and doing so substantially more efficiently. Additionally, we compare our results to heuristic-based approaches and show that these methods do not provide sufficient explanations in most cases, whereas the explanations of our approach are guaranteed to be sufficient. An additional advantage of our method is that it enables the *progressive convergence* of the refined explanations to the final explanation, as illustrated in Fig. 1. This approach allows practitioners to observe minimal subsets across various reduced networks, offering a fine-grained interpretation of the model's prediction. Additionally, it provides the possibility of halting the process once the explanation meets some desired criteria.

Besides these practical aspects, we view this work as a novel proof-of-concept for using abstraction-refinement-based techniques in explainability, obtaining formally provable explanations over abstract neural networks, which allow significantly more efficient verification and fine-grained interpretation over abstracted and refined networks. We hence consider this work a significant step in exploring explanations with formal guarantees for neural networks.

## 2. Preliminaries

### 2.1. Notation

We denote scalars with lower-case letters, vectors with bold lower-case letters, matrices with bold upper-case letters, and sets in calligraphic font. The $i$-th dimension of a vector $\boldsymbol{x}$ is denoted by $\boldsymbol{x}_{(i)}$. Given some $n \in \mathbb{N}$, let $[n] := \{1, \ldots, n\}$. Given a set $\mathcal{S} \subset \mathbb{R}^n$ and a function $f : \mathbb{R}^n \to \mathbb{R}^m$, we define $f(\mathcal{S}) := \{f(s) \mid s \in S\}$.

## 2.2. Neural Network Verification

We specify a generic neural network classifier architecture that can utilize any element-wise nonlinear activation function. For an input $\boldsymbol{x} \in \mathbb{R}^n$, the neural network classifier is denoted as $f \colon \mathbb{R}^n \to \mathbb{R}^c$. Numerous tools have been proposed for formally verifying properties of neural networks, with adversarial robustness being the most frequently examined property (Brix et al., 2023). The neural network verification query can be formalized as follows:

---

**Neural Network Verification (Problem Statement)**:
**Input**: A neural network $f$, such that $\boldsymbol{y} = f(\boldsymbol{x})$, with an input specification $\psi_{\text{in}}(\boldsymbol{x})$, and an unsafe output specification $\psi_{\text{out}}(\boldsymbol{y})$.
**Output**: *No*, if there exists some $\boldsymbol{x} \in \mathbb{R}^n$ such that $\psi_{\text{in}}(\boldsymbol{x})$ and $\psi_{\text{out}}(\boldsymbol{y})$ both hold, and *Yes* otherwise.

---

There exist many off-the-shelf neural network verifiers (Brix et al., 2023). If the input specifications $\psi_{in}(\boldsymbol{x})$, output specifications $\psi_{out}(\boldsymbol{y})$, and model $f$ are piecewise-linear (e.g., $f$ uses ReLU activations), this task can be solved exactly (Katz et al., 2017). However, the problem is often relaxed for efficiency, and the output is enclosed by bounding all approximation errors (see Sec. 6).

## 2.3. Formally Provable Minimal Sufficient Explanations

In this study, we concentrate on local post-hoc explanations for neural network classifiers. Specifically, for a neural network classifier $f \colon \mathbb{R}^n \to \mathbb{R}^c$ and a given local input $\boldsymbol{x} \in \mathbb{R}^n$ predicting class $t := \arg\max_j f(\boldsymbol{x})_{(j)} \in [c]$, our objective is to explain why $f$ classified $\boldsymbol{x}$ as class $t$. For regression tasks, a similar formulation can be found by limiting the deviation by some threshold $\delta \in \mathbb{R}_+$ (Appendix D).

**Sufficient Explanations.** A common method for interpreting the decisions of classifiers involves identifying subsets of input features $\mathcal{S} \subseteq [n]$ such that fixing these features to their specific values guarantees the prediction remains unchanged. Specifically, these techniques guarantee that the classification result remains consistent across *any* potential assignment within the complementary set $\bar{\mathcal{S}} := [n] \setminus \mathcal{S}$, thereby allowing for the formal verification of the explanations' soundness. While in the classic setting features in the complementary set $\bar{\mathcal{S}}$ are allowed to take on any possible feature values (Ignatiev et al., 2019; Darwiche & Hirth, 2020; Bassan & Katz, 2023), a more feasible and generalizable version restricts the possible assignments for $\bar{\mathcal{S}}$ to a bounded $\epsilon_p$-region (Wu et al., 2023; La Malfa et al., 2021; Izza et al., 2024). We use $(\boldsymbol{x}_{\mathcal{S}}; \tilde{\boldsymbol{x}}_{\bar{\mathcal{S}}}) \in \mathbb{R}^n$ to denote an assignment where the features of $\mathcal{S}$ are set to the values of the vector $\boldsymbol{x} \in \mathbb{R}^n$ and the features of $\bar{\mathcal{S}}$ are set to the values of another vector $\tilde{\boldsymbol{x}} \in \mathbb{R}^n$ within the $\epsilon_p$-region. Formally, we define a sufficient explanation $\mathcal{S}$ as follows:

**Definition 1** (Sufficient Explanation). *Given a neural network $f$, an input $\boldsymbol{x} \in \mathbb{R}^n$, a perturbation radius $\epsilon_p \in \mathbb{R}$, and a subset $\mathcal{S} \subseteq [n]$, we say that $\mathcal{S}$ is a sufficient explanation concerning the query $\langle f, \boldsymbol{x}, \mathcal{S}, \epsilon_p \rangle$ on an $\ell_p$-norm ball $B_p^{\epsilon_p}$ of radius $\epsilon_p \in \mathbb{R}_+$ around $\boldsymbol{x}$ iff it holds that:*

$$\forall \tilde{\boldsymbol{x}} \in B_p^{\epsilon_p}(\boldsymbol{x}) \colon [\arg\max_j \ f(\boldsymbol{x}_{\mathcal{S}}; \tilde{\boldsymbol{x}}_{\bar{\mathcal{S}}})_{(j)} = \arg\max_j \ f(\boldsymbol{x})_{(j)}],$$

$$\text{with} \quad B_p^{\epsilon_p}(\boldsymbol{x}) := \{\tilde{\boldsymbol{x}} \in \mathbb{R}^n \mid \|\boldsymbol{x} - \tilde{\boldsymbol{x}}\|_p \le \epsilon_p\}.$$

*We define* $\text{suff}(f, \boldsymbol{x}, \mathcal{S}, \epsilon_p) = 1$ *iff $\mathcal{S}$ constitutes a sufficient explanation with respect to the query $\langle f, \boldsymbol{x}, \mathcal{S}, \epsilon_p \rangle$, and* $\text{suff}(f, \boldsymbol{x}, \mathcal{S}, \epsilon_p) = 0$ *otherwise.*

Def. 1 can be formulated as a neural network verification query. This method has been proposed by prior studies, which employed these techniques to validate the sufficiency of specific subsets (Wu et al., 2023; Bassan & Katz, 2023; La Malfa et al., 2021; Izza et al., 2024).

**Minimal Explanations.** Clearly, if the subset $\mathcal{S}$ is chosen as the entire input set, i.e., $\mathcal{S} := [n]$, it is a sufficient explanation. However, a common view in the literature suggests that smaller subsets are more meaningful than larger ones (Ribeiro et al., 2018; Carter et al., 2019; Barceló et al., 2020; Ignatiev et al., 2019). Therefore, there is a focus on identifying subsets that not only are sufficient but also meet a criterion for minimality:

**Definition 2** (Minimal Sufficient Explanation). *Given a neural network $f$, an input $\boldsymbol{x} \in \mathbb{R}^n$, and a subset $\mathcal{S} \subseteq [n]$, we say that $\mathcal{S}$ is a minimal sufficient explanation concerning the query $\langle f, \boldsymbol{x}, \mathcal{S}, \epsilon_p \rangle$ on $B_p^{\epsilon_p}$ of radius $\epsilon_p$ iff $\mathcal{S}$ is a sufficient explanation, and for any $i \in \mathcal{S}$, $\mathcal{S} \setminus \{i\}$ is not a sufficient explanation. We define* $\text{min-suff}(f, \boldsymbol{x}, \mathcal{S}, \epsilon_p) = 1$ *if $\mathcal{S}$ satisfies both sufficiency and minimality concerning $\langle f, \boldsymbol{x}, \mathcal{S}, \epsilon_p \rangle$, and* $\text{min-suff}(f, \boldsymbol{x}, \mathcal{S}, \epsilon_p) = 0$ *otherwise.*

Minimal sufficient explanations can also be determined using neural network verifiers. Unlike simply verifying the sufficiency of a specific subset, this process requires executing multiple verification queries to ensure the minimality of the subset. Alg. 1 outlines such a procedure (similar methods are discussed in (Ignatiev et al., 2019; Wu et al., 2023; Bassan & Katz, 2023)). The algorithm begins with $\mathcal{S}$ encompassing the entire feature set $[n]$ and iteratively tries to exclude a feature $i$ from $\mathcal{S}$, each time checking whether $\mathcal{S} \setminus \{i\}$ remains sufficient. If $\mathcal{S} \setminus \{i\}$ is still sufficient, feature $i$ is removed; otherwise, it is retained in the explanation. This process is repeated until a minimal subset is obtained.

## 3. From Abstract Neural Networks to Abstract Explanations

A primary challenge in obtaining minimal sufficient explanations in neural networks is the high computational complexity involved, as verifying the sufficiency of a subset through

---

**Algorithm 1** Minimal Explanation Search

---

**Input:** Neural network $f \colon \mathbb{R}^n \to \mathbb{R}^c$, input $\boldsymbol{x} \in \mathbb{R}^n$, perturbation radius $\epsilon_p \in \mathbb{R}$

1: $\mathcal{S} \leftarrow [n]$
2: **for each** feature $i \in [n]$ **do**    ▷ suff$(f, \boldsymbol{x}, \mathcal{S}, \epsilon_p)$ holds
3:     **if** suff$(f, \boldsymbol{x}, \mathcal{S} \setminus \{i\}, \epsilon_p)$ **then**
4:        $\mathcal{S} \leftarrow \mathcal{S} \setminus \{i\}$
5:     **end if**
6: **end for**
7: **return** $\mathcal{S}$        ▷ min-suff$(f, \boldsymbol{x}, \mathcal{S}, \epsilon_p)$ holds

---

a neural network verification query is NP-Complete (Katz et al., 2017), making it especially difficult for larger networks (Brix et al., 2023). Obtaining a minimal subset also requires a linear number of verification queries relative to the input size (Alg. 1), making the process computationally intensive for large inputs. One potential solution is to replace the original neural network $f$ with a much smaller, abstract neural network $f'$, and then run verification queries on $f'$ instead of $f$. However, a key challenge here is to ensure that a sufficient explanation for $f'$ is also a sufficient explanation for $f$. Although similar ideas of such abstraction techniques have been applied to improve the efficiency of adversarial robustness verification (Elboher et al., 2020; Cohen et al., 2023; Ladner & Althoff, 2023; Liu et al., 2024), to our knowledge, we are the first to use an approach of this form to obtain provable explanations for neural networks.

**Abstract Neural Networks.** When abstracting a neural network, rather than using a traditional network $f \colon \mathbb{R}^n \to \mathbb{R}^c$, it is common to employ an abstract neural network $f'$ that outputs a set that encloses the actual output of $f$ (Ladner & Althoff, 2023; Prabhakar & Rahimi Afzal, 2019; Boudardara et al., 2022). This approach facilitates a more flexible propagation through the neural network, capturing the error due to the abstraction. More formally, we define the domain of our abstract network $f' \colon \mathbb{R}^n \to 2^{(\mathbb{R}^c)}$, where $2^{(\mathbb{R}^c)}$ denotes the power set of $\mathbb{R}^c$. In the simplest case, this means that $f'$ outputs a $c$-dimensional interval rather than a $c$-dimensional vector.

Since our abstract network $f'$ now outputs a set, we must define a sufficient explanation for an abstract network. Specifically, we define a sufficient explanation for $f'$ and a target class $t \in [c]$ as a subset $\mathcal{S} \subseteq [n]$ such that when the features in $\mathcal{S}$ are fixed to their values in $\boldsymbol{x}$, the lower bound for the target class $t$ consistently exceeds the upper bound of all other classes $j \neq t$:

**Definition 3** (Sufficient Explanation for Abstract Network).
$\mathcal{S} \subseteq [n]$ *is a sufficient explanation of an abstract network* $f'$ *concerning the query* $\langle f', \boldsymbol{x}, \mathcal{S}, \epsilon_p \rangle$ *iff* $\forall j \neq t \in [c]$:

$$\forall \tilde{\boldsymbol{x}} \in B_p^{\epsilon_p}(\boldsymbol{x}) \colon [\min(f'(\boldsymbol{x}_{\mathcal{S}}; \tilde{\boldsymbol{x}}_{\bar{\mathcal{S}}})_{(t)}) \geq \max(f'(\boldsymbol{x}_{\mathcal{S}}; \tilde{\boldsymbol{x}}_{\bar{\mathcal{S}}})_{(j)})],$$
$$\text{with } B_p^{\epsilon_p}(\boldsymbol{x}) \coloneqq \{\tilde{\boldsymbol{x}} \in \mathbb{R}^n \mid \|\boldsymbol{x} - \tilde{\boldsymbol{x}}\|_p \leq \epsilon_p\}.$$

**Neuron-Merging-Based Abstraction.** Various strategies can be employed to abstract a neural network to reduce its size. In this work, we abstract neural networks by merging neurons that exhibit similar behavior within the network for some bounded input set. For instance, numerous sigmoid neurons may become fully saturated, producing outputs close to 1. Hence, we can merge these saturated neurons and establish corresponding error bounds for the given input set. This can be realized without large computational overhead to a desired reduction rate $\rho \in [0, 1]$ such that the overall verification time, including abstraction, mainly depends on the remaining number of neurons. By using the construction suggested by (Ladner & Althoff, 2023), where merged neurons are replaced by a single one and the approximation error is bounded with set propagation, we can prove the important sufficiency implication property. We give details about the precise construction in Appendix A. For convenience, we define $\rho = 1$ as the original network.

We are now ready to establish the following claim on sufficient explanations for the query $\langle f', \boldsymbol{x}, \mathcal{S}, \epsilon_p \rangle$:

**Proposition 1** (Explanation Under Abstraction). *Given a neural network* $f$, *an input* $\boldsymbol{x}$, *a perturbation radius* $\epsilon_p$ *and a subset* $\mathcal{S} \subseteq [n]$, *let* $f'$ *be an abstract network constructed by neuron merging concerning the query* $\langle f, \boldsymbol{x}, \mathcal{S}, \epsilon_p \rangle$. *Then, it holds that:*

$$\mathrm{suff}(f', \boldsymbol{x}, \mathcal{S}, \epsilon_p) \implies \mathrm{suff}(f, \boldsymbol{x}, \mathcal{S}, \epsilon_p).$$

*Proof.* The proof can be found in Appendix A.2.    □

However, while a sufficient explanation $\mathcal{S}$ for the query $\langle f', \boldsymbol{x}, \mathcal{S}, \epsilon_p \rangle$ is also provably sufficient for the query $\langle f, \boldsymbol{x}, \mathcal{S}, \epsilon_p \rangle$, if $\mathcal{S}$ is insufficient for $\langle f', \boldsymbol{x}, \mathcal{S}, \epsilon_p \rangle$, it does not necessarily mean it is insufficient for $\langle f, \boldsymbol{x}, \mathcal{S}, \epsilon_p \rangle$. To more clearly highlight the explanation $\mathcal{S}$ within the context of the abstract network $f'$, we introduce an intermediate type of explanation, termed an *abstract sufficient explanation*. This is a provably sufficient explanation for $\langle f', \boldsymbol{x}, \mathcal{S}, \epsilon_p \rangle$ and, by extension (Prop. 1), also provably sufficient for $\langle f, \boldsymbol{x}, \mathcal{S}, \epsilon_p \rangle$:

**Definition 4** (Abstract Sufficient Explanation). *We define a sufficient explanation* $\mathcal{S}$ *concerning the query* $\langle f', \boldsymbol{x}, \mathcal{S}, \epsilon_p \rangle$ *as an abstract sufficient explanation concerning the query* $\langle f, \boldsymbol{x}, \mathcal{S}, \epsilon_p \rangle$.

**Running example.** We demonstrate these concepts using a toy ReLU network with positive weights and biases in Fig. 2, allowing for exact bounds via simplified interval-bound propagation. While intentionally simplified, the example serves to illustrate the procedure:

Fig. 2a shows the network and the interpreted input $(0, 1, 1)$. Biases are 0 except for the lower output neuron, which has a bias of 10. Propagating $(0, 1, 1)$ gives outputs of 15 (class

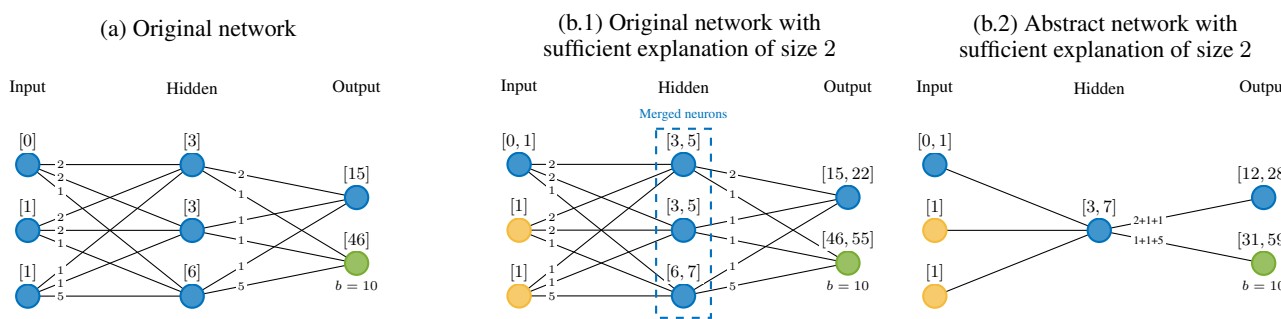

Figure 2: Running example. a) Original network with winner class 2 (green). b) Sufficient explanations (yellow) of original network (b.1) and abstract network (b.2).

1) and 46 (class 2), predicting class 2. Fig. 2b.1 shows an explanation that includes features 2 and 3 (yellow): We fix these features to their original values of 1, restricting their domains to $[1, 1]$. Feature 1 is not in the explanation and hence is allowed to vary freely within $[0, 1]$. We then compute bounds via interval propagation. For example, the top hidden neuron gets an input range of $[3, 5]$, from a lower bound of $0 \times 2 + 1 \times 2 + 1 \times 1 = 3$ and an upper bound of $1 \times 2 + 1 \times 2 + 1 \times 1 = 5$. These bounds are propagated to the output layer using the weights. For example, the top output neuron's range is $[15, 22]$, calculated as: lower bound is $3 \times 2 + 3 \times 1 + 6 \times 1 = 15$ and upper bound is $5 \times 2 + 5 \times 1 + 7 \times 1 = 22$. Overall, the output range for class 2 ($[46, 55]$) is strictly above that of class 1 ($[15, 22]$). Thus, fixing features 2 and 3 is *sufficient* to guarantee that class 2 remains the predicted class — making it a valid explanation.

In Fig. 2b.2, we show how three hidden neurons are merged by first unifying their intervals and then computing a weighted sum. While this process is not sound in general, we simplify the process here for clarity (See Appendix B.1 for the full example). For instance, the top neuron yields the interval $[12, 28]$ from $3 \cdot (2 + 1 + 1)$ and $7 \cdot (2 + 1 + 1)$. As this lies strictly below $[31, 59]$, features 2 and 3 form an *abstract explanation* per Def. 3.

Although we now have a framework that enables us to obtain explanations $\mathcal{S}$ for a neural network $f$ more efficiently — since queries on the smaller abstract network are faster — there is still an issue. The explanation $\mathcal{S}$ generated from the abstract network $f'$ might not be minimal for the original network $f$, even if it is minimal with respect to $\langle f', \boldsymbol{x}, \mathcal{S}, \epsilon_p \rangle$. In order to guarantee the minimality of the explanation in the original network, we apply refinement.

## 4. From Refining Neural Networks to Refining Explanations

To produce an explanation that is both sufficient and minimal, we apply an iterative refinement process. In each step, we construct a slightly larger refined network $f''$ than the

previously constructed abstract network $f'$ by splitting some of the merged neurons, resulting in a larger reduction rate $\rho'' > \rho'$. The refined abstract network $f''$ is still substantially smaller than the original network $f$ but slightly larger than $f'$, allowing us to generate a smaller explanation:

**Proposition 2** (Refined Abstract Network). *Given a neural network $f$, an input $\boldsymbol{x}$, a perturbation radius $\epsilon_p \in \mathbb{R}_+$, a subset $\mathcal{S} \subseteq [n]$, and an abstract network $f'$ with reduction rate $\rho' \in [0, 1]$, we can construct a refined abstract network $f''$ from $f, f'$ with reduction rate $\rho'' > \rho'$, for which holds:*

$$\forall \tilde{\boldsymbol{x}} \in B_p^{\epsilon_p}(\boldsymbol{x}) \colon f(\boldsymbol{x}_{\mathcal{S}}; \tilde{\boldsymbol{x}}_{\bar{\mathcal{S}}}) \in f''(\boldsymbol{x}_{\mathcal{S}}; \tilde{\boldsymbol{x}}_{\bar{\mathcal{S}}}) \subset f'(\boldsymbol{x}_{\mathcal{S}}; \tilde{\boldsymbol{x}}_{\bar{\mathcal{S}}}).$$

*Proof.* The proof can be found in Appendix A.3. $\square$

Considering a refined abstract network $f''$ in relation to $f$ and $f'$ with $\rho'' > \rho'$, the following property holds for the explanations generated for these networks:

**Proposition 3** (Intermediate Sufficient Explanation). *Let there be a neural network $f$, an abstract network $f'$, and a refined neural network $f''$. Then, it holds that:*

$$\text{suff}(f', \boldsymbol{x}, \mathcal{S}, \epsilon_p) \implies \text{suff}(f'', \boldsymbol{x}, \mathcal{S}, \epsilon_p) \text{ and}$$
$$\text{suff}(f'', \boldsymbol{x}, \mathcal{S}, \epsilon_p) \implies \text{suff}(f, \boldsymbol{x}, \mathcal{S}, \epsilon_p).$$

*Proof.* The proof can be found in Appendix A.4. $\square$

We observe that any sufficient explanation $\mathcal{S}$ for the query $\langle f'', \boldsymbol{x}, \mathcal{S}, \epsilon_p \rangle$ is also sufficient for the query $\langle f, \boldsymbol{x}, \mathcal{S}, \epsilon_p \rangle$ but might not be for $\langle f', \boldsymbol{x}, \mathcal{S}, \epsilon_p \rangle$ (Prop. 3). Thus, the intermediate explanation of a refined network is a subset of the explanation of the abstract network. Consequently, we define, in a manner akin to abstract sufficient explanations, an intermediate category termed *refined sufficient explanations*:

**Definition 5** (Refined Sufficient Explanation). *We define a sufficient explanation $\mathcal{S}$ concerning $\langle f'', \boldsymbol{x}, \mathcal{S}, \epsilon_p \rangle$ as a refined sufficient explanation, where the refined abstract network $f''$ is constructed with respect to $\langle f, f', \boldsymbol{x}, \mathcal{S}, \epsilon_p \rangle$ for a neural network $f$, and an abstract network $f'$.*

The observation that a sufficient explanation for $\langle f'', \boldsymbol{x}, \mathcal{S}, \epsilon_p \rangle$ is a subset of the one for $\langle f', \boldsymbol{x}, \mathcal{S}, \epsilon_p \rangle$

suggests the following characteristic about the minimality of sufficient explanations:

**Proposition 4** (Intermediate Minimal Sufficient Explanation). *Let there be a neural network $f$, an abstract network $f'$, and a refined neural network $f''$. Then, if $\mathcal{S}$ is a sufficient explanation concerning $f$, $f'$, and $f''$, it holds that:*

$$\text{min-suff}(f, \boldsymbol{x}, \mathcal{S}, \epsilon_p) \implies \text{min-suff}(f'', \boldsymbol{x}, \mathcal{S}, \epsilon_p) \text{ and}$$
$$\text{min-suff}(f'', \boldsymbol{x}, \mathcal{S}, \epsilon_p) \implies \text{min-suff}(f', \boldsymbol{x}, \mathcal{S}, \epsilon_p).$$

*Proof.* The proof can be found in Appendix A.5. □

We note that the implication chain in Prop. 4 is in reverse order compared to the implication chain in Prop. 3. Intuitively, refining the abstract network $f'$ to $f''$ incrementally produces larger neural networks, which in turn generates progressively smaller explanations, until ultimately converging to a minimal explanation for some refined network as it converges to the original network. We harness this iterative process and propose an abstraction-refinement approach to produce such minimal subsets in Alg. 2.

---

**Algorithm 2** Minimal Abstract Explanation Search

---

**Input:** Neural network $f \colon \mathbb{R}^n \to \mathbb{R}^c$, input $\boldsymbol{x} \in \mathbb{R}^n$, target $t = \arg\max_j f(x)_{(j)} \in [c]$, perturbation radius $\epsilon_p \in \mathbb{R}$

1: Init $\mathcal{S} \leftarrow [n]$, $\mathcal{F} \leftarrow [n]$
2: **for each** feature $i \in \mathcal{F}$ **do**      ▷ suff$(f, \boldsymbol{x}, \mathcal{S}, \epsilon_p)$ holds
3:     $f' \leftarrow$ Abstract $f$ w.r.t suff$(f, \boldsymbol{x}, \mathcal{S} \setminus \{i\}, \epsilon_p)$
4:     **do**
5:         **if** suff$(f', \boldsymbol{x}, \mathcal{S} \setminus \{i\}, \epsilon_p)$ **then**          ▷ Def. 3
                     ▷ Feature $i$ is *not* in minimal explanation
6:             $\mathcal{S} \leftarrow \mathcal{S} \setminus \{i\}$; **break**
7:         **else**
8:             $\tilde{\boldsymbol{x}} \leftarrow$ Obtain counterexample
                     w.r.t suff$(f', \boldsymbol{x}, \mathcal{S} \setminus \{i\}, \epsilon_p)$
9:             **if** $\arg\max_j f(\tilde{\boldsymbol{x}})_{(j)} \neq t$ **then**
                     ▷ Feature $i$ *must* be in explanation
10:                $\mathcal{F} \leftarrow \mathcal{F} \setminus \{i\}$; **break**
11:            **else**              ▷ Abstraction too coarse
12:                $f'' \leftarrow$ Refine $f'$          ▷ Prop. 2
13:                $f' \leftarrow f''$   ▷ Update abstract network
14:            **end if**
15:        **end if**
16:    **while** true          ▷ Repeat with refined network
17: **end for**
18: **return** $\mathcal{S}$          ▷ min-suff$(f, \boldsymbol{x}, \mathcal{S}, \epsilon_p)$ holds

---

The algorithm starts with a coarse abstract network $f'$ and derives an abstract sufficient explanation $\mathcal{S}$ by progressively removing features from $\mathcal{S}$, akin to the method described in Alg. 1. All following line numbers are with respect to Alg. 2. While the abstract sufficient explanation is provably sufficient for the original network (Prop. 3), it is not

necessarily provably minimal (Prop. 4). If we cannot be certain whether a subset $\mathcal{S}$ is sufficient for the abstract network (lines 7 to 15), we check whether feature $i$ is indeed not in the explanation by extracting a counterexample and checking its output in the original network (lines 8 to 9). If the counterexample is spurious due to the abstraction in $f'$, we refine the abstract neural network and thus produce a slightly larger network $f''$ (line 12). Using this refined network $f''$, we acquire a refined sufficient explanation relative to it, which allows us to remove additional features from the explanation as the abstraction error is smaller (Prop. 2). As we remove additional features from the explanation, the verification query to test whether the current subset is a sufficient explanation becomes harder as additional features can be perturbed. Thus, it is sensible to only abstract the network in line 3 to a level for which the verification query was still successful, i.e., use the reduction rate of $f''$. This process continues, with each iteration slightly enlarging the abstract network through refinement and consequently reducing the size of $\mathcal{S}$.

**Proposition 5** (Greedy Minimal Sufficient Explanation Search). *Alg. 2 produces a provably sufficient and minimal explanation $\mathcal{S}$ concerning the query $\langle f, \boldsymbol{x}, \mathcal{S}, \epsilon_p \rangle$, which converges to the same explanation as obtained by Alg. 1.*

*Proof.* The proof can be found in Appendix A.6. □

While Alg. 2 converges to the same explanation $\mathcal{S}$ as Alg. 1, it operates on smaller networks and thus returns a smaller explanation if a timeout is applied. In addition, one obtains a fine-grained interpretation of the network's prediction across different abstraction levels (Fig. 1).

**Complexity.** The runtime of Algorithm 2 is bounded by $\mathcal{O}((n + \xi) \cdot \max_t)$, where $n$ is the number of features, $\xi$ the number of refinement queries, and $\max_t$ the maximum time required to certify a single query. A higher value of $\xi$ corresponds to a more fine-grained refinement process, which typically reduces $\max_t$ by allowing more features to be handled through coarser abstractions. Conversely, smaller values of $\xi$ imply fewer refinement steps and thus longer certification queries — for instance, $\xi = 1$ corresponds to a single verification query over the full, unabstracted network.

**Running example (cont.).** We continue our running example from Fig. 2 in Fig. 3: Fig. 3c.1 show that feature 3 alone is also an explanation. While it is a *sufficient* explanation for the original network, it is not sufficient for the abstract network (Fig. 3c.2), since $[4, 28]$ and $[17, 59]$ overlap — violating Def. 3. This indicates that although every abstract explanation is valid for the original network, the reverse is not necessarily true. Consequently, *minimal* explanations for the abstract network, while sufficient, may not remain minimal when applied to the original network.

Fig. 3d shows a refinement merging only the first two neu-

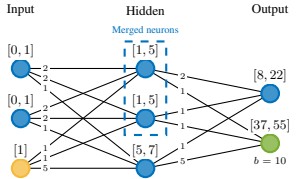

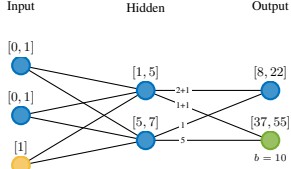

Figure 3: Running example (cont.). c) Minimal sufficient explanation for the original network (c.1) which is insufficient for the abstract network (c.2). d) Refining the abstract network results in a sufficent explanation.

rons, yielding one merged neuron for them and another for the third. The output interval $[37, 55]$ remains strictly above $[8, 22]$, confirming that fixing only feature 3 is a *sufficient* explanation for both the *refined and original network*.

## 5. Experimental Results

**Experimental Setup.** We implemented the algorithms using CORA (Althoff, 2015) as the backend neural network verifier. We performed our experiments on three image classification benchmarks: (a) MNIST (LeCun, 1998), (b) CIFAR-10 (Krizhevsky & Hinton, 2009), and (c) GT-SRB (Stallkamp et al., 2012). Comprehensive details about the models and their training are provided in Appendix B, and additional experiments, ablation studies, and extensions to other domains are provided in Appendix C and D.

### 5.1. Comparison to Standard Verification-Based Explanations

In our initial experiment, we aimed to evaluate the abstraction-refinement method proposed in Alg. 2 against the traditional approach described in Alg. 1 for deriving provably sufficient explanations for neural networks, as implemented in previous studies (see (Wu et al., 2023; Bassan & Katz, 2023; La Malfa et al., 2021)). Complete details about the implementation of the refinement process are available in Appendix B. We assessed the effectiveness of both approaches using the two most prevalent metrics for evaluating sufficient explanations, as documented in (Wu et al., 2023; Ignatiev et al., 2019; Bassan & Katz, 2023): (i) the size of the explanation, with smaller sizes indicating higher meaningfulness, and (ii) the computation time.

Fig. 4 shows the explanation size over time for each benchmark. We observed that the abstraction-refinement approach significantly outperforms the standard greedy method in computation time ($-41\%$ for MNIST, $-36\%$ for CIFAR-10, $-56\%$ for GTSRB). We also implemented a timeout for each dataset and assessed the explanation size for each method under the timeout. These results are presented in Tab. 1 and demonstrate the substantial improvements in explanation size achieved with our abstraction-refinement approach.

Table 1: Mean explanation size with a timeout of 100s, $10^3$s, and $10^4$s for MNIST, CIFAR-10, and GTSRB, respectively.

| | **Explanation size** | | |
|---|---|---|---|
| **Method** | **MNIST** | **CIFAR-10** | **GTSRB** |
| Ours | $\mathbf{204.4}\pm129.2$ | $\mathbf{308.2}\pm236.6$ | $\mathbf{230.6}\pm234.3$ |
| Standard | $408.7\pm36.6$ | $448.7\pm138.4$ | $502.4\pm101.7$ |
| $\rho = 10\%$ | $507.3\pm141.3$ | $850.6\pm28.9$ | $502.4\pm101.7$ |
| $\rho = 20\%$ | $420.6\pm149.4$ | $806.4\pm57.4$ | $704.6\pm168.8$ |
| $\rho = 30\%$ | $340.0\pm145.0$ | $687.1\pm130.8$ | $605.0\pm203.1$ |
| $\rho = 50\%$ | $285.5\pm98.0$ | $346.3\pm219.4$ | $392.6\pm240.1$ |
| $\rho = 70\%$ | $325.6\pm64.5$ | $311.5\pm233.0$ | $292.6\pm225.2$ |
| $\rho = 90\%$ | $372.7\pm46.8$ | $310.6\pm233.1$ | $333.7\pm188.4$ |

### 5.2. Explanations at Different Abstraction Levels

Besides improving computation time and reducing explanation size, the abstraction-refinement method allows users to observe the progressive decrease in explanation size at each abstraction level. Although networks with significant reductions initially provide larger explanations, refining these networks obtains explanations of decreasing size. These explanations at different abstraction levels may provide users with deeper insights and transparency into the prediction mechanism. Furthermore, it offers the flexibility to halt the process when the explanation is provably sufficient, even if not provably minimal. This fine-grained process is illustrated in Fig. 5, where small explanations can be obtained with low reduction rates ($\rho \leq 50\%$ of the neurons).

### 5.3. Effect of Reduction Rates

For a more detailed analysis of our findings, we present additional results on the computation of explanations at varying reduction rates within our abstraction process. In Fig. 6, we illustrate the percentage of processed features verified to be included or excluded from the explanation per reduction rate for MNIST, CIFAR-10, and GTSRB. These results highlight that the majority of the explanation processing occurs at coarser abstractions, i.e., smaller network sizes $\rho$, resulting in improved computation time.

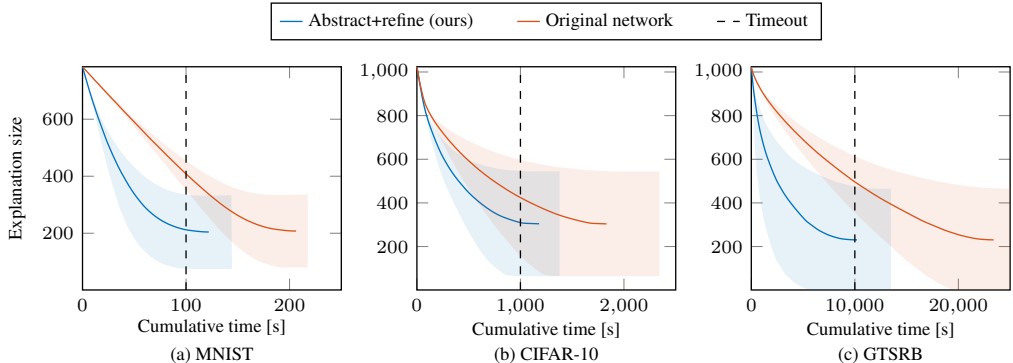

Figure 4: The explanation size over cumulative time for MNIST, CIFAR10, and GTSRB, throughout the entire abstraction-refinement algorithm, or using the standard verification algorithm on the original network. The standard deviation is shown as a shaded region.

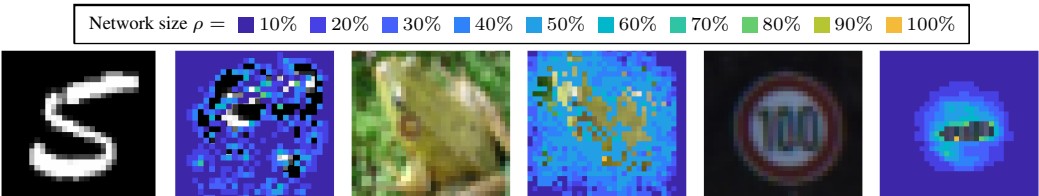

Figure 5: Examples of explanations at varying reduction rates for MNIST, CIFAR-10, and GTSRB.

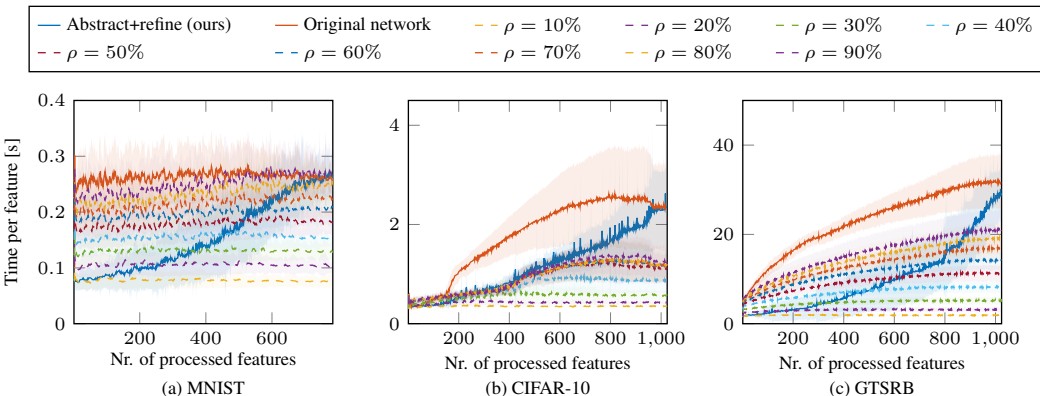

Figure 6: The percentage of processed features—either included or excluded from the explanation—over cumulative time, segmented by reduction rate, throughout the abstraction-refinement algorithm, or using the standard verification algorithm on the original network.

Table 2: Comparison with heuristic-based approaches, measuring sufficiency and average computation time over 100 images. Heuristic methods are faster but lack sufficiency, while our method consistently provides sufficient explanations (Prop. 5).

| | MNIST | | CIFAR-10 | | GTSRB | |
|---|---|---|---|---|---|---|
| **Method** | Suff. | Time | Suff. | Time | Suff. | Time |
| Anchors (Ribeiro et al., 2018) | 25% | 0.56s | 3% | 1.17s | 17% | 0.14s |
| SIS (Carter et al., 2019) | 22% | 322.72s | 0% | 553.92s | 6% | 95.13s |
| Original network | **100**% | 207.80s | **100**% | 1,838.2s | **100**% | 23,504s |
| Abstract+refine (ours) | **100**% | 121.95s | **100**% | 1,180.5s | **100**% | 10,235s |

## 5.4. Comparison to Heuristic-Based Approaches

In our final experiment, we compare our explanations with two widely used non-verification-based methods for sufficient explanations: (i) Anchors (Ribeiro et al., 2018) and (ii) SIS (Carter et al., 2019). Although these methods operate relatively efficiently, they do not formally verify the sufficiency of the explanations, relying instead on a sampling heuristic across the complement. We depicted the comparisons between our verified explanations and those generated by Anchors and SIS in Tab. 2. These results highlight that while faster, these methods produce far fewer sufficient explanations ($\leq 25\%$).

## 6. Related Work

**Formal XAI.** Our work is closely related to *formal XAI* (Marques-Silva, 2023), which aims to provide explanations with formal guarantees. Previous research has focused on deriving provable sufficient explanations for decision trees (Huang et al., 2021; Izza et al., 2022; Bounia & Koriche, 2023; Arenas et al., 2022), linear models (Marques-Silva et al., 2020; Subercaseaux et al., 2025), monotonic classifiers (Marques-Silva et al., 2021; El Harzli et al., 2022), and tree ensembles (Izza & Marques-Silva, 2021; Ignatiev et al., 2022; Boumazouza et al., 2021; Audemard et al., 2022; 2023; Bassan et al., 2025a). More closely related to our work are methods that derive minimal sufficient explanations for neural networks (Bassan & Katz, 2023; La Malfa et al., 2021; Wu et al., 2023; Bassan et al., 2023; Izza et al., 2024; Azzolin et al., 2025), which often rely on verification tools. While these tools have improved in scalability (Wu et al., 2024; Wang et al., 2021b; Brix et al., 2023), computing provable explanations remains computationally expensive (Barceló et al., 2020; Wäldchen et al., 2021; Adolfi et al., 2025; Bassan et al., 2024; Ordyniak et al., 2023; Marzouk & De La Higuera, 2024; Marzouk et al., 2025; Amir et al., 2024; Huang et al., 2023) and typically requires multiple verification queries (Ignatiev et al., 2019). Other approaches mitigate these computational challenges by redefining sufficiency (Jin et al., 2025; Izza et al., 2023; Wang et al., 2021a; Chockler et al., 2024; Jin et al., 2025), applying smoothing (Xue et al., 2023), or using self-explaining methods (Bassan et al., 2025b; Alvarez Melis & Jaakkola, 2018; Bassan et al., 2025c).

**Abstraction-refinement.** Our algorithm leverages abstraction-refinement, a technique commonly used to enhance the efficiency of symbolic model checking (Clarke et al., 2000; Wang et al., 2006), and applied in software (Jhala & Majumdar, 2009; Flanagan & Qadeer, 2002), hardware (Andraus et al., 2007), and hybrid systems verification (Alur et al., 2000). Recently, abstraction-refinement has been used to improve the efficiency of certifying adversarial robustness by abstracting neural network sizes (Elboher

et al., 2020; 2022; Ladner & Althoff, 2023; Liu et al., 2024; Siddiqui et al., 2024). However, to the best of our knowledge, our work is the first to use an abstraction-refinement technique to reduce neural network sizes for obtaining provable explanations of neural network predictions. Lastly, while similar ideas of model pruning leveraging probabilistic or uncertainty concepts have improved scalability in feature attribution generation (e.g., (Ancona et al., 2019)), our non-probabilistic method offers stronger guarantees on continuous domains. Additionally, our novel refinement method ensures provable guarantees of minimality.

## 7. Limitations and Future Work

A limitation of our framework, as with any method solving this task, is its dependence on neural network verification queries. Verification techniques, although still currently limited for state-of-the-art models, are rapidly advancing in scalability (Wang et al., 2021b; Brix et al., 2023). Our method adds an orthogonal step in using these tools to derive provable explanations for neural network decisions. Hence, as the scalability of verification tools improves, so will that of our approach. Importantly, compared to methods addressing the same task, our approach is significantly more efficient and handles much larger models. Moreover, while we focus on minimal sufficient explanations, future work could extend our abstraction-refinement strategies to other provable explanations. To demonstrate the broader applicability of our method, we discuss several extensions in Appendix D. These include: (i) *regression* tasks, (ii) expansions into *language* domains, and (iii) various enhancements to our algorithm, such as different feature orderings and certain scalability-sufficiency trade-offs.

## 8. Conclusion

Obtaining minimal sufficient explanations for neural networks offers a promising way to provide explanations with formally verifiable guarantees. However, the scalability of generating such explanations is hindered by the need to invoke multiple neural network verification queries. Our abstraction-refinement approach addresses this by starting with a significantly smaller network and refining it as needed. This ensures that the explanations are provably sufficient for the original network and, ultimately, both provably minimal and sufficient. Our method also produces intermediate explanations, allowing for an efficient early stop when sufficient but non-minimal explanations are reached, while also offering a more fine-grained interpretation of the model prediction across abstractions. Our experiments demonstrate that our approach generates minimal sufficient explanations substantially more efficiently than traditional methods, representing a significant step forward in producing explanations for neural network predictions with formal guarantees.

## Impact Statement

Since our work contributes to advancing both the practical and theoretical dimensions of explainability, it encounters implications common to other frameworks in the field, including susceptibility to adversarial attacks, privacy concerns, oversimplifications, and potential biases. By prioritizing explanations with provable guarantees, we strive to improve trustworthiness. However, as the primary focus of our work lies on systematic and methodological aspects, we believe it does not have any immediate or direct social ramifications.

## Acknowledgements

This work was partially funded by the European Union (ERC, VeriDeL, 101112713). Views and opinions expressed are however those of the author(s) only and do not necessarily reflect those of the European Union or the European Research Council Executive Agency. Neither the European Union nor the granting authority can be held responsible for them. This research was additionally supported by a grant from the Israeli Science Foundation (grant number 558/24) and from the project FAI (No. 286525601) funded by the German Research Foundation (Deutsche Forschungsgemeinschaft, DFG).

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

# Appendix

The appendix includes supplementary experimental results, implementation details, and proofs.

**Appendix A** contains all proofs of this paper along with further details on the abstraction method.
**Appendix B** contains implementation details.
**Appendix C** contains additional experiments and ablation studies.
**Appendix D** discusses the extension to other domains than image classification, including regression and language tasks.

## A. Proofs

In this section, we provide the missing proofs in the same order as they appear in the main paper.

### A.1. Preliminary Definitions and Lemmas

To prove Prop. 1, we define both a neural network and an abstract neural network layer-by-layer, as described in Sec. 3.

**Definition 6** (Neural Network). *Let $\boldsymbol{x} \in \mathbb{R}^{n_0}$ be the input of a neural network $f$ with $\kappa$ layers, its output $\boldsymbol{y} := f(\boldsymbol{x}) \in \mathbb{R}^{n_\kappa}$ is obtained as follows:*

$$\boldsymbol{h}_0 := \boldsymbol{x}, \; \boldsymbol{h}_k := L_k\left(\boldsymbol{h}_{k-1}\right), \; \boldsymbol{y} = \boldsymbol{h}_\kappa, \quad k \in [\kappa],$$

*where $L_k \colon \mathbb{R}^{n_{k-1}} \to \mathbb{R}^{n_k}$ represents the operation of layer $k$ and is given by $L_k\left(\boldsymbol{h}_{k-1}\right) := \sigma(\boldsymbol{W}_k \boldsymbol{h}_{k-1} + \boldsymbol{b}_k)$ with weight matrix $\boldsymbol{W}_k \in \mathbb{R}^{n_k \times n_{k-1}}$, bias $\boldsymbol{b}_k \in \mathbb{R}^{n_k}$, activation function $\sigma \colon \mathbb{R}^{n_k} \to \mathbb{R}^{n_k}$, and number of neurons $n_k \in \mathbb{N}$.*

An abstract network is then described by:

**Definition 7** (Abstract Neural Network). *Let $\boldsymbol{x} \in \mathbb{R}^{n_0}$ be the input of an abstract neural network $f'$ with $\kappa$ layers, its output $\boldsymbol{y} := f'(\boldsymbol{x}) \subset \mathbb{R}^{n_\kappa}$ is obtained as follows:*

$$\mathcal{H}'_0 := \{\boldsymbol{x}\}, \; \mathcal{H}'_k := L'_k\left(\mathcal{H}'_{k-1}\right), \; \boldsymbol{y} = \mathcal{H}'_\kappa, \quad k \in [\kappa],$$

*where $L'_k \colon 2^{(\mathbb{R}^{n'_{k-1}})} \to 2^{(\mathbb{R}^{n'_k})}$ represents the operation of the abstract layer $k$ and is given by $L'_k\left(\mathcal{H}'_{k-1}\right) = \sigma(\boldsymbol{W}'_k \mathcal{H}'_{k-1} \oplus \boldsymbol{b}'_k)$ with weight matrix $\boldsymbol{W}'_k \in \mathbb{R}^{n'_k \times n'_{k-1}}$, bias $\boldsymbol{b}'_k \in \mathbb{R}^{n'_k}$, activation function $\sigma \colon \mathbb{R}^{n'_k} \to \mathbb{R}^{n'_k}$, number of neurons $n'_k \in \mathbb{N}$, $n'_0 := n_0, n'_\kappa := n_\kappa$, and $\oplus$ denoting the Minkowski sum (Given two sets $\mathcal{S}_1, \mathcal{S}_2 \subset \mathbb{R}^n$, then $\mathcal{S}_1 \oplus \mathcal{S}_2 = \{s_1 + s_2 \mid s_1 \in \mathcal{S}_1, \; s_2 \in \mathcal{S}_2\}$).*

Let $\mathcal{X} \subset \mathbb{R}^n$ be the set of points satisfying the input specification $\psi_{\text{in}}(\boldsymbol{x})$ for a point $\boldsymbol{x} \in \mathbb{R}^n$ (Sec. 2.2). As mentioned in Sec. 2.2, the exact output $\mathcal{Y}^* := f(\mathcal{X})$ is often infeasible to compute, and an enclosure $\mathcal{Y} \supset \mathcal{Y}^*$ is computed instead by bounding all approximation errors. This is realized by iteratively propagating $\mathcal{X}$ through all layers and enclosing the output of each layer. For example, given an input set $\mathcal{H}_{k-1} \subset \mathcal{H}^*_{k-1}$ to layer $k$, we obtain the output $\mathcal{H}_k \supset L_k\left(\mathcal{H}_{k-1}\right) = \mathcal{H}^*_k$, with $\mathcal{H}^*_0 = \mathcal{H}_0 = \mathcal{X}$ and $\mathcal{Y} = \mathcal{H}_\kappa$. Let $\mathcal{H}_k, \mathcal{Y}$ and $\mathcal{H}'_k, \mathcal{Y}'$ denote the enclosures of the sets $\mathcal{H}^*_k, \mathcal{Y}_k$ using the original network $f$ and the abstract network $f'$, respectively. In this work, we use a neuron-merging construction defined by (Ladner & Althoff, 2023):

**Lemma 1** (Neuron-Merging Construction (Ladner & Althoff, 2023, Prop. 4)). *Given a layer $k \in [\kappa - 1]$ of a network $f$, output bounds $\mathcal{I}_k \subset \mathbb{R}^{n_k}$, a set of neurons to merge $\mathcal{B}_k \subseteq [n_k]$, and the indices of the remaining neurons $\overline{\mathcal{B}}_k := [n_k] \backslash \mathcal{B}_k$, the layer $k$ and $k + 1$ are constructed as follows:*

$$\boldsymbol{W}'_k := \boldsymbol{W}_{k(\overline{\mathcal{B}}_k, \cdot)}, \; \boldsymbol{b}'_k := \boldsymbol{b}_{k(\overline{\mathcal{B}}_k)}, \quad \boldsymbol{W}'_{k+1} = \boldsymbol{W}_{k+1(\cdot, \overline{\mathcal{B}}_k)}, \; \boldsymbol{b}'_{k+1} = \boldsymbol{b}_{k+1} \oplus \boldsymbol{W}_{k+1(\cdot, \mathcal{B}_k)} \mathcal{I}_{k(\mathcal{B}_k)},$$

*where $\boldsymbol{W}_{k+1(\cdot, \mathcal{B}_k)} \mathcal{I}_{k(\mathcal{B}_k)}$ is the approximation error. The construction ensures that $\mathcal{H}^*_{k+1} \subseteq \mathcal{H}'_{k+1}$.*

Given a neural network $f$, an input $\boldsymbol{x}$, a perturbation radius $\epsilon_p$, and a subset $\mathcal{S} \subseteq [n]$, we say that $f'$ is an abstract network constructed by neuron merging with respect to the query $\langle f, \boldsymbol{x}, \mathcal{S}, \epsilon_p \rangle$ if we define the input set $\mathcal{X} := \mathcal{B}^{\epsilon_p}_p(\boldsymbol{x}_\mathcal{S}; \tilde{\boldsymbol{x}}_{\overline{\mathcal{S}}})$ and recursively apply the neuron-merging construction as described in Lemma 1 for any two layers $L_{k-1}$ and $L_k$. We can now provide an explicit proof of Prop. 1:

### A.2. Proof of Prop. 1

**Proposition 1** (Explanation Under Abstraction)**.** *Given a neural network $f$, an input $\boldsymbol{x}$, a perturbation radius $\epsilon_p$ and a subset $\mathcal{S} \subseteq [n]$, let $f'$ be an abstract network constructed by neuron merging concerning the query $\langle f, \boldsymbol{x}, \mathcal{S}, \epsilon_p \rangle$. Then, it holds that:*

$$\mathrm{suff}(f', \boldsymbol{x}, \mathcal{S}, \epsilon_p) \implies \mathrm{suff}(f, \boldsymbol{x}, \mathcal{S}, \epsilon_p).$$

*Proof.* We prove this statement by contradiction: Assume that $\mathcal{S}$ is a sufficient explanation for the abstract network, i.e., for $\langle f', \boldsymbol{x}, \epsilon_p \rangle$, but not for the original network, i.e., for $\langle f, \boldsymbol{x}, \epsilon_p \rangle$. Given that $\mathcal{S}$ is a sufficient explanation for $\langle f', \boldsymbol{x}, \epsilon_p \rangle$, the following holds (Def. 3):

$$\forall j \neq t \in [c], \ \forall \tilde{\boldsymbol{x}} \in B_p^{\epsilon_p}(\boldsymbol{x}): \quad [\min(f'(\boldsymbol{x}_{\mathcal{S}}; \tilde{\boldsymbol{x}}_{\bar{\mathcal{S}}})_{(t)}) \geq \max(f'(\boldsymbol{x}_{\mathcal{S}}; \tilde{\boldsymbol{x}}_{\bar{\mathcal{S}}})_{(j)})], \tag{1}$$

where $t := \arg\max_j \ f(\boldsymbol{x})_{(j)}$ is the target class. Moreover, since $\mathcal{S}$ is *not* a sufficient explanation concerning $\langle f, \boldsymbol{x}, \epsilon_p \rangle$ it follows that (Def. 1):

$$\exists \tilde{\boldsymbol{x}}' \in B_p^{\epsilon_p}(\boldsymbol{x}): \quad [\arg\max_j \ f(\boldsymbol{x}_{\mathcal{S}}; \tilde{\boldsymbol{x}}'_{\bar{\mathcal{S}}})_{(j)} \neq \arg\max_j \ f(\boldsymbol{x})_{(j)} = t]. \tag{2}$$

Since Eq. (1) is valid for *any* $\tilde{\boldsymbol{x}} \in B_p^{\epsilon_p}(\boldsymbol{x})$, it explicitly applies to $\tilde{\boldsymbol{x}}' \in B_p^{\epsilon_p}(\boldsymbol{x})$ as well. Specifically, we have:

$$\forall j \in [c] \backslash \{t\}, \quad [\min(f'(\boldsymbol{x}_{\mathcal{S}}; \tilde{\boldsymbol{x}}'_{\bar{\mathcal{S}}})_{(t)}) \geq \max(f'(\boldsymbol{x}_{\mathcal{S}}; \tilde{\boldsymbol{x}}'_{\bar{\mathcal{S}}})_{(j)})]. \tag{3}$$

We now assert that to establish the correctness of the proposition, it suffices to demonstrate that:

$$f(\boldsymbol{x}_{\mathcal{S}}; \tilde{\boldsymbol{x}}'_{\bar{\mathcal{S}}}) \in f'(\boldsymbol{x}_{\mathcal{S}}; \tilde{\boldsymbol{x}}'_{\bar{\mathcal{S}}}) \tag{4}$$

The rationale is as follows: if Eq. (4) holds, it would directly contradict our initial assumption. To begin, observe that Eq. (4) directly leads to:

$$\forall j \in [c], \quad [\min(f'(\boldsymbol{x}_{\mathcal{S}}; \tilde{\boldsymbol{x}}'_{\bar{\mathcal{S}}})_{(j)}) \leq f(\boldsymbol{x}_{\mathcal{S}}; \tilde{\boldsymbol{x}}'_{\bar{\mathcal{S}}})_{(j)} \leq \max(f'(\boldsymbol{x}_{\mathcal{S}}; \tilde{\boldsymbol{x}}'_{\bar{\mathcal{S}}})_{(j)})], \tag{5}$$

and therefore, based on Eq. (3), it will directly follow that:

$$\forall j \in [c] \backslash \{t\}, \quad [f(\boldsymbol{x}_{\mathcal{S}}; \tilde{\boldsymbol{x}}'_{\bar{\mathcal{S}}})_{(t)} \geq \min(f'(\boldsymbol{x}_{\mathcal{S}}; \tilde{\boldsymbol{x}}'_{\bar{\mathcal{S}}})_{(t)}) \geq \max(f'(\boldsymbol{x}_{\mathcal{S}}; \tilde{\boldsymbol{x}}'_{\bar{\mathcal{S}}})_{(j)}) \geq f(\boldsymbol{x}_{\mathcal{S}}; \tilde{\boldsymbol{x}}'_{\bar{\mathcal{S}}})_{(j)}]. \tag{6}$$

Now, specifically, given that the following condition is satisfied:

$$\forall j \neq t \in [c], \quad [f(\boldsymbol{x}_{\mathcal{S}}; \tilde{\boldsymbol{x}}'_{\bar{\mathcal{S}}})_{(t)} \geq f(\boldsymbol{x}_{\mathcal{S}}; \tilde{\boldsymbol{x}}'_{\bar{\mathcal{S}}})_{(j)}]. \tag{7}$$

This indicates that $\arg\max_j \ f(\boldsymbol{x}_{\mathcal{S}}; \tilde{\boldsymbol{x}}'_{\bar{\mathcal{S}}})(j) = t$, which contradicts the assumption stated in Eq. (2).

We are now left to prove that $f(\boldsymbol{x}_{\mathcal{S}}; \tilde{\boldsymbol{x}}'_{\bar{\mathcal{S}}}) \in f'(\boldsymbol{x}_{\mathcal{S}}; \tilde{\boldsymbol{x}}'_{\bar{\mathcal{S}}})$. Let $\boldsymbol{h}_k$ and $\mathcal{H}'_k$ be as defined in Def. 6 and Def. 7, respectively, for the input $(\boldsymbol{x}_{\mathcal{S}}; \tilde{\boldsymbol{x}}'_{\bar{\mathcal{S}}}) \in \mathcal{X}$, where $\mathcal{X} := B_p^{\epsilon_p}(\boldsymbol{x}_{\mathcal{S}}; \tilde{\boldsymbol{x}}'_{\bar{\mathcal{S}}})$. Recall that this is defined since the merging is performed with respect to the query $\langle f, \boldsymbol{x}, \mathcal{S}, \epsilon_p \rangle$. We show by induction that the statement $f(\boldsymbol{x}_{\mathcal{S}}; \tilde{\boldsymbol{x}}'_{\bar{\mathcal{S}}}) \in f'(\boldsymbol{x}_{\mathcal{S}}; \tilde{\boldsymbol{x}}'_{\bar{\mathcal{S}}})$ holds:

*Induction hypothesis.* $k \in [\kappa]$: The condition $\boldsymbol{h}_k \in \mathcal{H}'_k$ is satisfied if a neuron merging operation was performed between any two layers up to and including layer $k - 1$.

*Induction base.* $k = 0$: Trivially holds (Def. 7).

*Induction step.* $k \to k + 1$: We need to show that $\boldsymbol{h}_{k+1} \in \mathcal{H}'_{k+1}$. Let $\mathcal{B}_k, \mathcal{I}_k$ be as in Lemma 1 and $\mathcal{H}'_k$ the output set of layer $k$ before merging. Thus, $\mathcal{I}_k \supset \mathcal{H}'_k$ holds. From the induction hypothesis, we know that $\boldsymbol{h}_k \in \mathcal{H}'_k$ holds. Recall from

Def. 6 that $\boldsymbol{h}_{k+1} = \sigma(\boldsymbol{W}_{k+1}\boldsymbol{h}_k + \boldsymbol{b}_{k+1})$. Doing the same for $\mathcal{H}'_k$ (Def. 7) and applying the neuron merging construction (Lemma 1) gives us:

$$\boldsymbol{h}_{k+1} \in \sigma(\boldsymbol{W}_{k+1}\mathcal{H}'_k \oplus \boldsymbol{b}_{k+1}) = \sigma(\boldsymbol{W}_{k+1(\cdot,\mathcal{B}_k)}\mathcal{H}'_{k(\mathcal{B}_k)} \oplus \boldsymbol{W}_{k+1(\cdot,\overline{\mathcal{B}}_k)}\mathcal{H}'_{k(\overline{\mathcal{B}}_k)} \oplus \boldsymbol{b}_{k+1})$$
$$\subseteq \sigma(\boldsymbol{W}_{k+1(\cdot,\mathcal{B}_k)}\mathcal{I}_{k(\mathcal{B}_k)} \oplus \boldsymbol{W}_{k+1(\cdot,\overline{\mathcal{B}}_k)}\mathcal{H}'_{k(\overline{\mathcal{B}}_k)} \oplus \boldsymbol{b}_{k+1})$$
$$= \mathcal{H}'_{k+1},$$

which proves the induction step. As the induction hypothesis holds for $k = \kappa$, we conclude that $f'(\boldsymbol{x}_\mathcal{S}; \tilde{\boldsymbol{x}}'_{\bar{\mathcal{S}}}) \in f'(\boldsymbol{x}_\mathcal{S}; \tilde{\boldsymbol{x}}'_{\bar{\mathcal{S}}})$ must be true. This, as previously explained, implies that $\arg\max_j\ f(\boldsymbol{x}_\mathcal{S}; \tilde{\boldsymbol{x}}'_{\bar{\mathcal{S}}})_{(j)} = t$, which contradicts our assumption in Eq. (2). $\qquad\square$

## A.3. Proof of Prop. 2

**Proposition 2** (Refined Abstract Network). *Given a neural network $f$, an input $\boldsymbol{x}$, a perturbation radius $\epsilon_p \in \mathbb{R}_+$, a subset $\mathcal{S} \subseteq [n]$, and an abstract network $f'$ with reduction rate $\rho' \in [0,1]$, we can construct a refined abstract network $f''$ from $f, f'$ with reduction rate $\rho'' > \rho'$, for which holds:*

$$\forall \tilde{\boldsymbol{x}} \in B_p^{\epsilon_p}(\boldsymbol{x})\colon\ f(\boldsymbol{x}_\mathcal{S}; \tilde{\boldsymbol{x}}_{\bar{\mathcal{S}}}) \in f''(\boldsymbol{x}_\mathcal{S}; \tilde{\boldsymbol{x}}_{\bar{\mathcal{S}}}) \subset f'(\boldsymbol{x}_\mathcal{S}; \tilde{\boldsymbol{x}}_{\bar{\mathcal{S}}}).$$

*Proof.* The containment of $f(\boldsymbol{x}_\mathcal{S}; \tilde{\boldsymbol{x}}_{\bar{\mathcal{S}}})$ follows using an analogous proof as for Prop. 1. While the subset relation does not hold in general when applying the abstraction (Lemma 1) using $\rho''$ instead of $\rho'$ as different neurons might be merged, one can restrict the neurons that are allowed to be merged to the subset of neurons $\mathcal{N}' = \cup_{k \in [\kappa-1]}\mathcal{B}_k$ that were merged to obtain $f'$. Using this restriction and as $\rho'' > \rho'$ holds, $\mathcal{N}'' \subset \mathcal{N}'$ holds. Then, as all additionally merged neurons $\mathcal{N}' \setminus \mathcal{N}''$ in $f'$ induce outer approximations and everything else is equal, the subset relation holds. $\qquad\square$

## A.4. Proof of Prop. 3

**Proposition 3** (Intermediate Sufficient Explanation). *Let there be a neural network $f$, an abstract network $f'$, and a refined neural network $f''$. Then, it holds that:*

$$\text{suff}(f', \boldsymbol{x}, \mathcal{S}, \epsilon_p) \implies \text{suff}(f'', \boldsymbol{x}, \mathcal{S}, \epsilon_p) \text{ and}$$
$$\text{suff}(f'', \boldsymbol{x}, \mathcal{S}, \epsilon_p) \implies \text{suff}(f, \boldsymbol{x}, \mathcal{S}, \epsilon_p).$$

*Proof.* We show the first implication by contradiction: Let us assume that $\mathcal{S}$ is a sufficient explanation for $f'$ but not for $f''$. Thus, the query $\langle f'', \boldsymbol{x}, \mathcal{S}, \epsilon_p \rangle$ does not fulfill Def. 3. However, as $f''(\boldsymbol{x}_\mathcal{S}; \tilde{\boldsymbol{x}}_{\bar{\mathcal{S}}}) \subset f'(\boldsymbol{x}_\mathcal{S}; \tilde{\boldsymbol{x}}_{\bar{\mathcal{S}}})$ holds due to Prop. 2, the query $\langle f', \boldsymbol{x}, \mathcal{S}, \epsilon_p \rangle$ can also not fulfill Def. 3, which contradicts our assumption. The proof for the second implication is analogous. $\qquad\square$

## A.5. Proof of Prop. 4

**Proposition 4** (Intermediate Minimal Sufficient Explanation). *Let there be a neural network $f$, an abstract network $f'$, and a refined neural network $f''$. Then, if $\mathcal{S}$ is a sufficient explanation concerning $f$, $f'$, and $f''$, it holds that:*

$$\text{min-suff}(f, \boldsymbol{x}, \mathcal{S}, \epsilon_p) \implies \text{min-suff}(f'', \boldsymbol{x}, \mathcal{S}, \epsilon_p) \text{ and}$$
$$\text{min-suff}(f'', \boldsymbol{x}, \mathcal{S}, \epsilon_p) \implies \text{min-suff}(f', \boldsymbol{x}, \mathcal{S}, \epsilon_p).$$

*Proof.* The statement is shown by contradiction for the relation of $f$ and $f''$: Let us assume that the explanation $\mathcal{S}$ is minimal for $f$ but not for $f''$. Thus, there must be a $\mathcal{S}' \subset \mathcal{S}$ which is minimal for $f''$. However, this cannot be an explanation for $f$ as $\mathcal{S}$ is already minimal for $f$, From Prop. 1 it follows that $\mathcal{S}'$ is also a minimal explanation for $f$, which contradicts our assumption that $\mathcal{S}$ is minimal. Analogous reasoning holds for $f''$ and $f'$. $\qquad\square$

## A.6. Proof of Prop. 5

**Proposition 5** (Greedy Minimal Sufficient Explanation Search). *Alg. 2 produces a provably sufficient and minimal explanation $\mathcal{S}$ concerning the query $\langle f, \boldsymbol{x}, \mathcal{S}, \epsilon_p \rangle$, which converges to the same explanation as obtained by Alg. 1.*

*Proof.* All line numbers are with respect to Alg. 2: The invariant described in line 2 holds due to Prop. 3. Thus, the final explanation $\mathcal{S}$ is sufficient concerning the original network $f$.

We show the minimality by contradiction: Let us assume the final explanation $\mathcal{S}$ is not minimal: There exists a feature $i \in [n]$ such that $\mathcal{S} \setminus \{i\}$ is still sufficient (Def. 2). It follows that we cannot have found a counterexample in line 8, Thus, to break the do-while loop, the sufficiency check in line 5 must have passed either on an abstract network or eventually after all refinement steps on the original network. However, this would remove feature $i$ from the explanation, which violates our assumption.

We converge to the same explanation as Alg. 1, as we process the features in the same order, Prop. 3 holds, and if removing a feature results in a non-sufficient explanation and no counterexample on the original network can be found, we refine the abstract networks until we converge to the original network. □

## B. Implementation Details

In this section, we offer further technical details about the implementation of our algorithms and provide specifics on the model architectures and training methods used in this study.

### B.1. Details on Abstract Explanations

In the running example in Sec. 3 and Sec. 4, we use a simplified process for merging neurons to illustrate the abstraction for clarity. However, as discussed in both Sec. 3, Sec. 4, and Appendix A, the actual method employs the Minkowski sum (Lemma 1), which is sound and obtains a tight bound while preserving the abstraction properties. For completeness, we provide here a simple version of the running example that reflects the use of this bound.

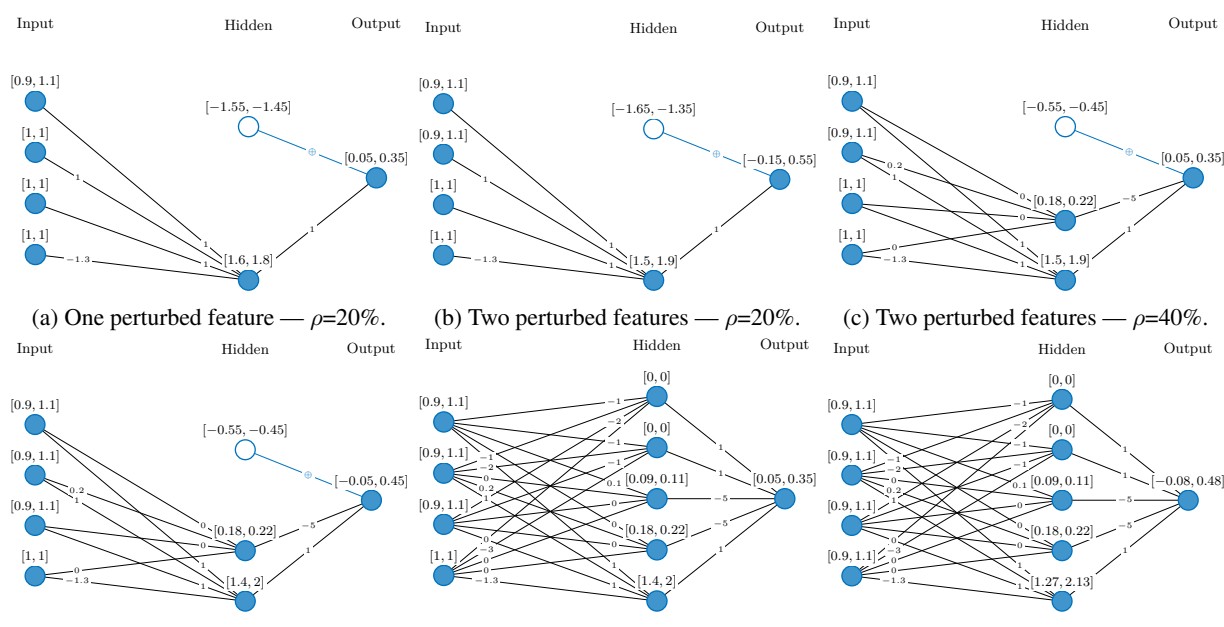

(a) One perturbed feature — $\rho$=20%.  (b) Two perturbed features — $\rho$=20%.  (c) Two perturbed features — $\rho$=40%.

(d) Three perturbed features — $\rho$=40%.  (e) Three perturbed features — $\rho$=100%.  (f) Four perturbed features — $\rho$=100%.

Figure 7: An example of a full abstraction refinement procedure using the tighter Minkowski sum instead of the weighted sum.

Fig. 7 illustrates, similar to the running example in Sec. 3 and Sec. 4, the specified bounds for each input feature and hidden ReLU activation neuron. We consider a binary classification task where the output is interpreted as class 1 if the output neuron has a positive value, and class 0 otherwise. For the initial input $[1, 1, 1, 1]$, the output is positive, resulting in a classification of 1. When neurons are merged at a rate of $\rho = 40\%$, an explanation of size 2 is sufficient; however, the same explanation size is inadequate when $\rho = 20\%$. For the full network ($\rho = 100\%$), the cardinality-minimal explanation has a size of 1.

Table 3: Dimensions for the MNIST classifier.

| Layer type | Paramater | Activation |
|---|---|---|
| Input | $784 \times 200$ | Sigmoid |
| Fully-connected | $200 \times 200$ | Sigmoid |
| Fully-connected | $200 \times 200$ | Sigmoid |
| Fully-connected | $200 \times 200$ | Sigmoid |
| Fully-connected | $200 \times 200$ | Sigmoid |
| Fully-connected | $200 \times 200$ | Sigmoid |
| Fully-connected | $200 \times 200$ | Sigmoid |
| Fully-connected | $200 \times 10$ | Softmax |

Table 4: Dimensions for the CIFAR-10 classifier.

| Layer type | Paramater | Activation |
|---|---|---|
| Input | $32 \times 32 \times 3$ | ReLU |
| Convolution | $32 \times 3 \times 4 \times 4$ | ReLU |
| Convolution | $64 \times 32 \times 4 \times 4$ | ReLU |
| Fully-connected | $32768 \times 128$ | ReLU |
| Fully-connected | $128 \times 64$ | ReLU |
| Fully-connected | $64 \times 10$ | Softmax |

Table 5: Dimensions for the GTSRB classifier.

| Layer type | Paramater | Activation |
|---|---|---|
| Input | $32 \times 32 \times 3$ | Sigmoid |
| Convolution | $16 \times 3 \times 3 \times 3$ | Sigmoid |
| Avg.-pooling | $2 \times 2$ | — |
| Convolution | $32 \times 16 \times 3 \times 3$ | Sigmoid |
| Avg.-pooling | $2 \times 2$ | — |
| Fully-connected | $4608 \times 128$ | Sigmoid |
| Fully-connected | $128 \times 43$ | softmax |

### B.2. Implementation details of Alg. 1 and Alg. 2

Both Alg. 1 and Alg. 2 iterate over the features in a specified order. This approach aligns with the methodologies used in (Wu et al., 2023; Izza et al., 2024; Bassan & Katz, 2023), where a sensitivity traversal over the features is employed. We prioritize iterating over features with the lowest sensitivity first, as they are most likely to be successfully freed, thus leading to a smaller explanation. As we refine the abstract network following Prop. 2, we define a series of reduction rates used during each refinement step. For simplicity, we start with a coarsest abstraction at a reduction rate $\rho = 10\%$ of the original network's neurons and increase $\rho$ by $10\%$ at each subsequent refinement, until $\rho = 100\%$ is reached, which restores the original network.

We also note that while MNIST utilizes grayscale images, both CIFAR-10 and GTSRB use RGB images. Following standard practices for colored images (Wu et al., 2023; Ribeiro et al., 2018; Carter et al., 2019; 2021), we provide explanations for CIFAR-10 and GTSRB on a per-pixel basis, rather than at the neuron level; this means we either include/exclude all color channels of a pixel within the explanation or none. Consequently, the maximum size of an explanation, $|\mathcal{S}|$, is $32 \cdot 32$ instead of $32 \cdot 32 \cdot 3$. For MNIST, which has only one color channel, the maximum size is always $|\mathcal{S}| = 28 \cdot 28$.

### B.3. Training and Model Implementation

For MNIST and CIFAR-10, we utilized common models from the neural network verification competition (VNN-COMP) (Brix et al., 2023), which are frequently used in experiments related to neural network verification tasks. Specifically, the MNIST model architecture is sourced from the ERAN benchmark within VNN-COMP, and the CIFAR-10 model is derived from the "marabou" benchmark. Since GTSRB is not directly utilized in VNN-COMP, we trained this model using a batch size of 32 for 10 epochs with the ADAM optimizer, achieving an accuracy of $84.8\%$. The precise dimensions and configurations of the models used for both VNN-COMP (MNIST and CIFAR-10) and GTSRB are provided: Tab. 3 for MNIST, Tab. 4, and Tab. 5 for GTSRB. For MNIST and GTSRB, we use a perturbation radius $\epsilon_\infty = 0.01$ as commonly used in VNN-COMP benchmarks, and for CIFAR-10, we use a smaller perturbation radius $\epsilon_\infty = 0.001$ as we have found this network to be not very robust.

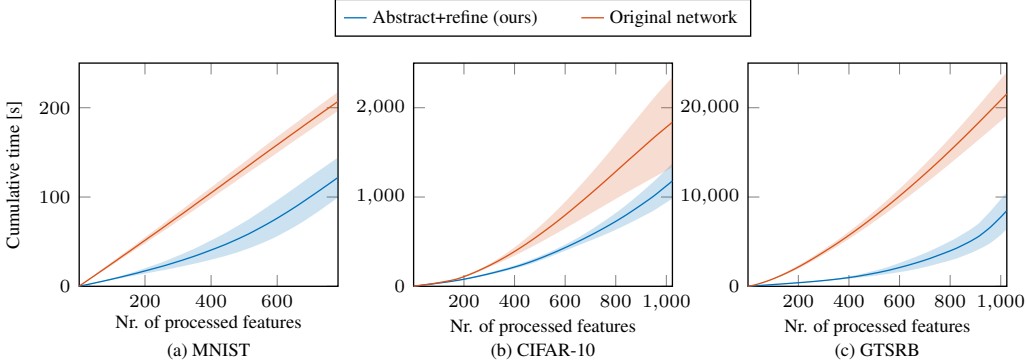

Figure 8: The percentage of features successfully processed—identified as either included or excluded from the explanation—over cumulative time for MNIST, CIFAR10, and GTSRB, throughout the entire abstraction-refinement algorithm, or using the standard verification algorithm on the original network. The standard deviation is shown as a shaded region.

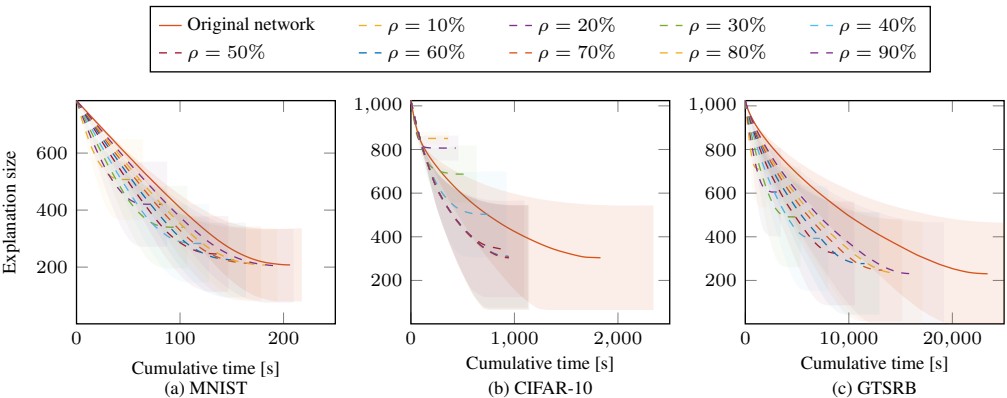

Figure 9: The explanation size over cumulative time for MNIST, CIFAR10, and GTSRB, segmented by reduction rate, throughout the entire abstraction-refinement algorithm, or using the standard verification algorithm on the original network. The standard deviation is shown as a shaded region.

## C. Additional experiments and ablation studies

### C.1. Additional experiments on image classification

In this section, we present further experimental results to complement those discussed in the main body of the paper. We begin by expanding on Fig. 4, which illustrates the change in explanation size over time for the standard verification method versus the abstraction-refinement approach. In Fig. 8, we offer a similar comparison, this time focusing on the number of processed features, i.e., features that have been selected to be included or excluded from the explanation. It is evident that the abstraction-refinement method processes features more efficiently than the standard approach, leading to enhanced scalability.

We continue to build on the findings presented in Fig. 6, which illustrates the number of features processed at various reduction rates. In Fig. 9, we similarly demonstrate the change in explanation size over time across different reduction rates. As lower reduction rates $\rho$ initially have a much steeper curve than larger ones, the explanation size is reduced faster. However, these lower reduction rates converge to larger explanation sizes than larger reduction rates. Our approach benefits from both worlds by initially using the steepest curve to reduce the explanation size, and automatically switching to the next steeper curve if no features can be freed anymore using the current rate.

Lastly, we provide some qualitative figures that depict the iterative abstraction-refinement process in the explanations across different reduction rates. Fig. 10 displays the initial images paired with a colored grid, where each color represents a specific reduction rate. These images are selected from the MNIST, CIFAR10, and GTSRB datasets. In the last row, we show some

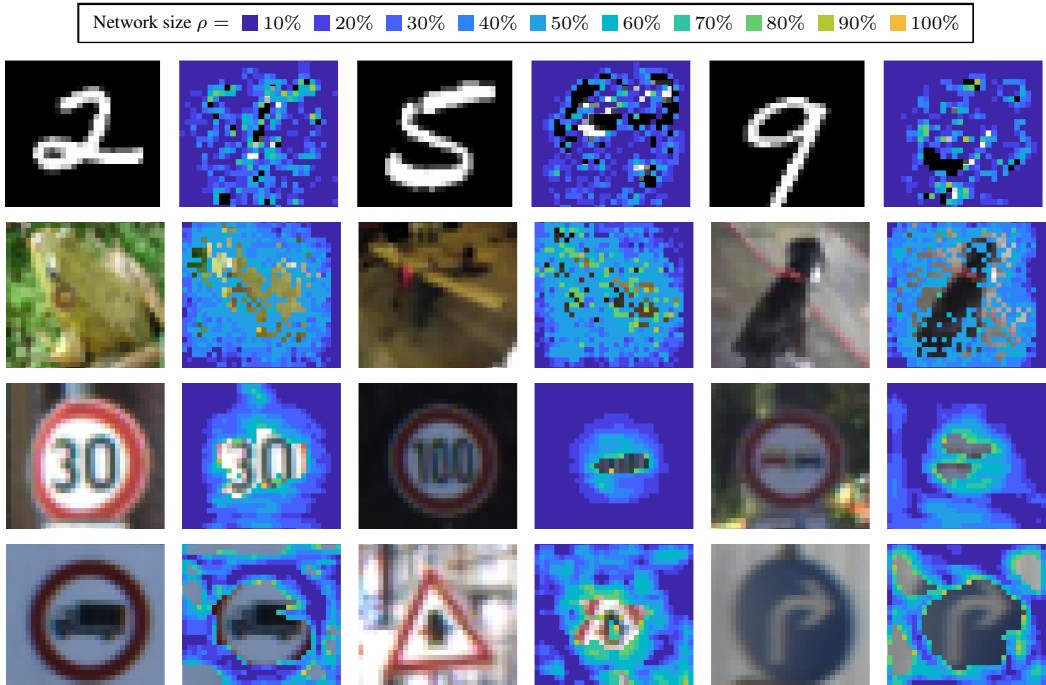

Figure 10: Original images compared to images featuring the complete abstraction-refinement grid at various abstraction rates for MNIST, CIFAR10, and GTSRB.

explanations of GTSRB images with unexpected explanations. For example, for the first image in the last row, the red circle surrounding the sign does not seem to be very important, as these pixels could be removed from the explanations using the coarsest abstraction ($\rho = 10\%$).

Additionally, to provide a more detailed visualization of the entire abstraction-refinement explanation process, which allows users to halt the verification at any stage, we include visualizations of both abstract and refined explanations at various steps and reduction rates. These visualizations are shown for all three benchmarks — MNIST, CIFAR-10, and GTSRB — in Fig. 11.

Finally, we also want to give a visual comparison of our approach and heuristic approaches in Fig. 12: First, images Fig. 12b-d highlight the importance of minimality. For example, the interior of the forward sign is not included in our explanation, and the edges alone are sufficient for classification. This shows the interior pixels are irrelevant and can be excluded without affecting the prediction. In contrast, non-minimal sufficient subsets include unnecessary features. To better grasp the significance of small or minimal explanations, one can consider the extreme case where the entire image is chosen as the explanation: although it is clearly sufficient, it is neither minimal nor informative. Secondly, image Fig. 12f-h shows that fixing the provably sufficient subset ensures robustness: any change in the complement within the domain does not alter the classification. This contrasts with heuristic subsets, where changes in the complement can flip the prediction – revealing their limitations. The explanation also matches human intuition, as the highlighted region alone justifies the predicted class, unlike the much less clear heuristic explanations.

## C.2. Ablation Study

### C.2.1. SUFFICIENCY-COMPUTATION TIME TRADE-OFF

In this subsection, we will examine the impact of varying perturbation radii $\epsilon_p$ on our experimental results. Larger $\epsilon_p$ perturbations make each query more challenging but provide stronger sufficiency guarantees. However, as the sufficiency conditions become more stringent, the total number of queries decreases, leading to larger explanation sizes. We conducted an experiment on MNIST using different perturbation radii, with the results presented in Tab. 6.

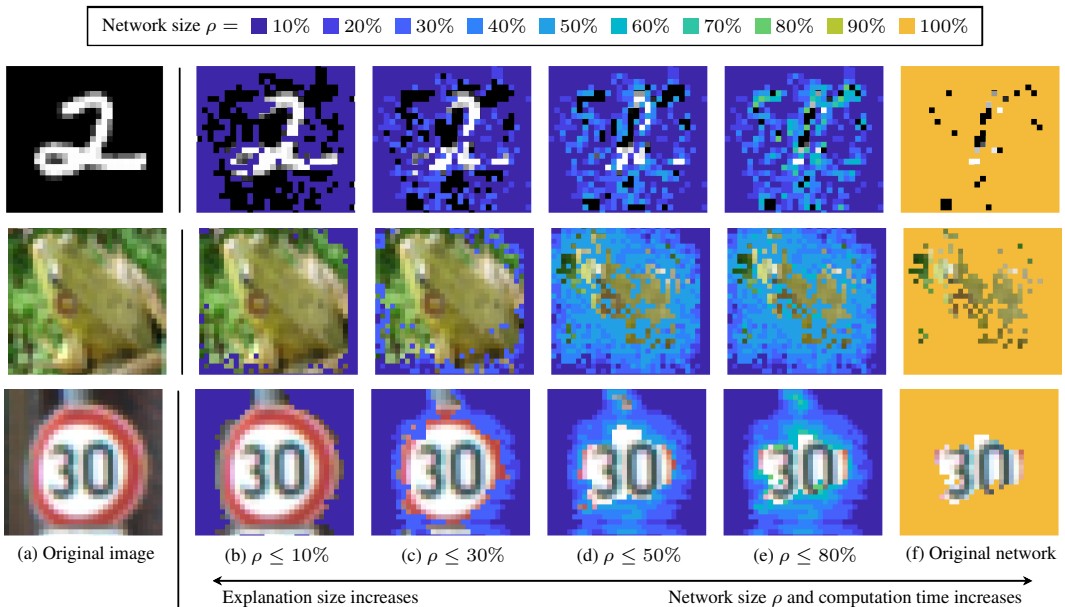

Figure 11: A step-by-step visualization of the different abstraction levels for both the network and explanation across MNIST, CIFAR-10, and GTSRB.

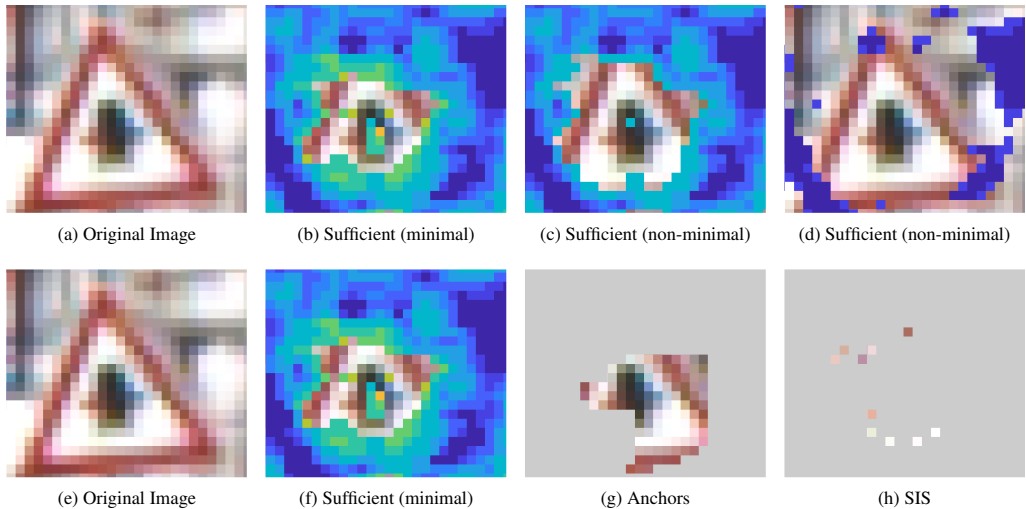

Figure 12: Visual comparisons of sufficiency and minimality of explanations.

Table 6: Impact of perturbation radius $\epsilon_p$ on explanation size and computation time.

| Method | Perturbation radius | Explanation size | Computation time [s] |
|---|---|---|---|
|  | 0.012 | 219.450±142.228 | 110.111±33.712 |
|  | 0.011 | 186.970±140.435 | 101.881±41.625 |
| Abstract+refine (ours) | 0.010 | 153.240±135.733 | 94.897±46.244 |
|  | 0.009 | 119.040±127.271 | 81.889±52.578 |
|  | 0.008 | 87.530±113.824 | 62.607±58.084 |
|  | 0.007 | 59.420±95.607 | 53.072±56.709 |

Table 7: Impact of feature order on explanation size and computation time.

| Method | Feature order | Explanation size | Computation time [s] |
|---|---|---|---|
| Abstract+refine (ours) | Sensitivity | 153.24±135.73 | 90.26±44.54 |
| | Shapley | 175.70±150.09 | 93.10±45.39 |
| | In-order | 231.46±160.73 | 98.10±46.42 |

Table 8: Average time to solve the verification query for different reduction rates $\rho$.

| Abstraction | MNIST | CIFAR | GTSRB |
|---|---|---|---|
| $\rho = 0.1$ | 0.08 | 0.35 | 1.89 |
| $\rho = 0.2$ | 0.11 | 0.42 | 3.05 |
| $\rho = 0.3$ | 0.13 | 0.55 | 4.73 |
| $\rho = 0.4$ | 0.16 | 0.75 | 7.05 |
| $\rho = 0.5$ | 0.18 | 0.90 | 9.27 |
| $\rho = 0.6$ | 0.20 | 0.92 | 11.28 |
| $\rho = 0.7$ | 0.22 | 0.93 | 12.97 |
| $\rho = 0.8$ | 0.23 | 0.92 | 14.38 |
| $\rho = 0.9$ | 0.25 | 0.93 | 15.68 |

### C.2.2. FEATURE ORDERING

We illustrate the impact of different feature orderings on the explanations generated by our method. While we adopt the approach proposed by (Wu et al., 2023), which orders features by descending sensitivity values, we also present results for explanation sizes and computation times using alternative feature orderings in our MNIST configuration. These alternatives include ordering by descending Shapley value attributions (Lundberg & Lee, 2017) and, for comparison, a straightforward in-order traversal that results in larger subsets. The results are summarized in Tab. 7.

### C.2.3. LOCALIZATION OF CLASS-DISCRIMATIVE FEATURES

In this experiment, we evaluate how effectively sufficient explanations identify class-discriminative features — that is, the proportion of features included in the explanation that actually correspond to meaningful class indicators (e.g., an object within an image). Since GTSRB includes object detection annotations, we leveraged its ground truth to measure how many of the features highlighted by our method fall within the annotated object regions. We compared this to explanations generated by Anchors. Our method achieved 93.33% alignment with the annotated regions, significantly outperforming Anchors, which reached only 60.36%. This suggests that our explanations are more tightly focused around the object of interest in the image.

### C.2.4. AVERAGE VERIFICATION QUERY TIME

To provide further insights into the different components of our approach, we provide average time to solve the verification query for different reduction rates $\rho$ and over all benchmarks in Tab. 8.

### C.3. Choice of Activation Function

In our main experiments, we used networks with either ReLU or sigmoid activation, respectively (Appendix B). While this networks are taken from VNN-COMP (Brix et al., 2023), we provide a full picture in this study including both activation functions for both networks in Tab. 9.

### C.4. Analyzing different network sizes

While the previous experiment already show the scalability of our approach on different network sizes (Appendix B), the networks are on different datasets. In this study, we show results comparing three networks taken from the Marabou benchmark of VNN-COMP (Brix et al., 2023), which all have CIFAR-10 images as input (Tab. 10).

Table 9: Computation time and explanation size for varying activation functions.

| Dataset | Activation | Explanation size | | |
| --- | --- | --- | --- | --- |
| | | Timeout [s] | Standard | Ours |
| MNIST | ReLU | 20 | 439.74 | **380.36** |
| | Sigmoid | 100 | 408.7 | **204.4** |
| CIFAR-10 | ReLU | 1,000 | 448.7 | **308.2** |
| | Sigmoid | 1,000 | 576.50 | **309.13** |
| GTSRB | ReLU | 2,000 | 849.4 | **701.0** |
| | Sigmoid | 10,000 | 502.4 | **101.7** |

Table 10: Computation time of explanation for networks taken from the Marabou benchmark.

| Netowrk | Standard | Ours |
| --- | --- | --- |
| Small | 471.50±161.34 | **335.07**±47.33 |
| Medium | 1045.13±95.05 | **780.95**±18.01 |
| Large | 1953.34±509.34 | **1382.95**±161.49 |

## D. Extension to Additional Domains

Although our method primarily targets classification tasks in image domains, it is model-type agnostic. Furthermore, it can be easily adapted to regression tasks by defining the sufficiency conditions for a subset $\mathcal{S}$ for a model $f : \mathbb{R}^n \to \mathbb{R}$ and some given input $\boldsymbol{x} \in \mathbb{R}^n$ as:

$$\forall \tilde{\boldsymbol{x}} \in B_p^{\epsilon_p}(\boldsymbol{x}) \colon \quad \| f(\boldsymbol{x}_{\mathcal{S}}; \tilde{\boldsymbol{x}}_{\bar{\mathcal{S}}}) - f(\boldsymbol{x}) \|_p \leq \delta, \quad \delta \in \mathbb{R}_+. \tag{8}$$

**Comparison to results over Taxinet** (Wu et al., 2023): We aimed to compare our results over regression tasks to those conducted by (Wu et al., 2023) which ran a "traditional" computation of a provably sufficient explanation for neural networks over the Taxinet benchmark, which is a real-world safety-critical airborne navigation system (Julian et al., 2020). The authors of (Wu et al., 2023) obtain minimal sufficient explanations over three different benchmarks of varying sizes, two of which are relatively small, and one which is larger (the CNN architecture). We performed experiments using this architecture. Our abstraction refinement approach obtained explanations within $35.71 \pm 3.71$ seconds and obtained explanations of size $699.30 \pm 169.34$, which provides a substantial improvement over the results reported by (Wu et al., 2023) (8814.85 seconds, and explanation size was not reported). We additionally provide visualizations for some of our obtained explanations (Fig. 13 and 14).

**Extension to language tasks.** We present results from experiments conducted on the safeNLP benchmark (Casadio et al., 2025), trained on the medical safety NLP dataset sourced from the annual neural network verification competition

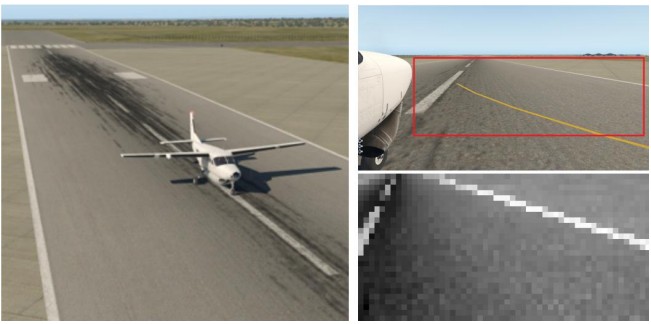

Figure 13: An autonomous aircraft taxiing scenario (Julian et al., 2020, Fig. 1), where images captured by a camera mounted on the right wing are cropped (red box) and downsampled

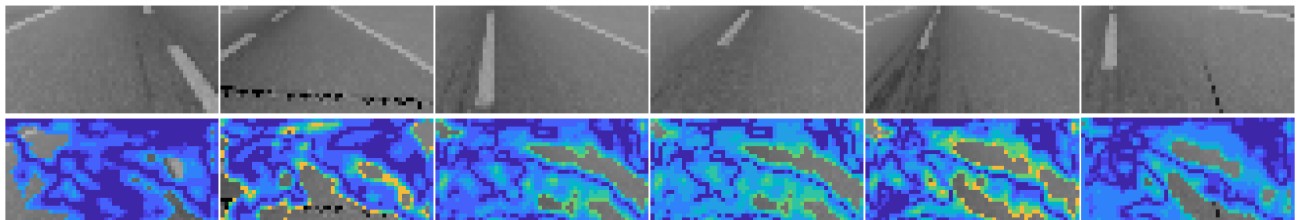

Figure 14: Varying results of explanations across different abstraction levels for the Taxinet benchmark.

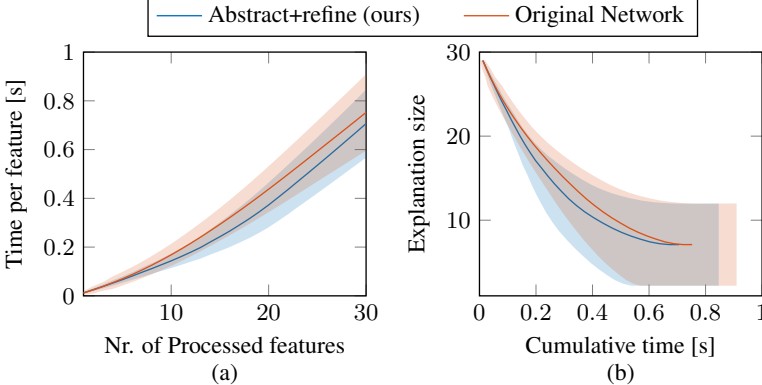

Figure 15: (a) The percentage of features successfully processed—identified as either included or excluded from the explanation—over cumulative time and (b) the explanation size over cumulative time for safeNLP, throughout the entire abstraction-refinement algorithm, or using the standard verification algorithm on the original network. The standard deviation is shown as a shaded region.

(VNN-COMP) (Brix et al., 2023). Notably, this benchmark is the only language-domain dataset included in the competition. The $\epsilon$ perturbations are applied within a latent space that represents an embedding of the input, thereby ensuring that the perturbations preserve the meaning of the sentence. Our findings are as follows: the traditional (non-abstraction-refinement) approach executed in $0.71 \pm 0.24$ seconds with an explanation size of $6.67 \pm 5.06$, while the abstraction-refinement approach completed in $0.66 \pm 0.22$ seconds, achieving the same explanation size of $6.67 \pm 5.06$. The results are also visualized in Fig. 15. The performance improvement here is relatively modest, as the benchmark contains few nonlinear activations. However, as emphasized in our study and the experimental analysis, the benefits of the abstraction-refinement method become significantly more pronounced in larger models with more nonlinear activations, and this benchmark should purely demonstrate that our approach can also be applied to non-image domains.

