# OpenReview forum: "Explaining, Fast and Slow: Abstraction and Refinement of Provable Explanations"
_ICML.cc/2025/Conference — ICML 2025 poster_

### Official Review · Reviewer_ZvCk · 2025-03-08

**Overall Recommendation:** 3

**Summary:**

The paper tackles the issue of computing verifiably sufficient input-level
explanations of neural network predictions.  The proposed algorithm speeds up
the computation, which involves several invokations to an exact solver, by
constructing a smaller-scale abstraction of the target neural network. The
abstraction retains sufficiency but may break minimality.  An incremental
version of the algorithm thus constructs a sequence of smaller-to-larger
abstractions until the corresponding explanation is both sufficient and
minimal.  The experiments primarily focus on evaluating efficiency and size
gains.

**Claims And Evidence:**

Evidence is quite convincing.

The main claims are that the explanations output by the proposed algorithm are
sufficient by construction and small, while the algorithm itself is claimed
to be faster than the competition.

- Sufficiency: this is supported both by the analysis of Algorithm 2 in page 6
  and by the results in Table 2: the explanations always achieve 100%
  sufficiency; the metric used for this assessment appears to be in line with
  the general notion of sufficiency in the formal explainability literature.

- Size: there appear to be sizeable (!) gains in terms of explanation size
  in Table 1.

- Speed: the results in Figure 2 indicate a reasonable speed-up on all data sets.

More generally, the algorithm is tested on three data sets and different neural
net architectures.  While the networks are quite small, this is due to
scalability issues of current NN verification tools, not directly the proposed
algorithm.

**Essential References Not Discussed:**

No to my knowledge.

**Experimental Designs Or Analyses:**

Yes, the design is clean.

**Methods And Evaluation Criteria:**

Yes.  The setup is very much in line with what's expected from an NN
verification paper (which slightly differs from what XAI researchers might
expect, but this difference is also expected).

**Other Comments Or Suggestions:**

- Algorithm 2, line 8: "obtain counterexample" - how is this done?

- Section 5: it is not immediately obvious what the "standard algorithm" is or
  whether it is reflective of the SOTA.  Please be clearer about it.

- Table 1: explanation size is not monotonic across $\rho$'s - this is a bit
  counterintuitive when compared with the discussion in Section 3.  Could
  you please clarify what is happening here?

- Figures and Tables: how is the std deviation computed?

- Proofs: it's not necessary to mention that proofs are in Appendix A four times.

- Prop 4: min-suff is technically undefined.

- Prop 2: the notation f \in f and f \subset f is not defined.

- It is really unfortunate that the construction by Ladner & Althoff is not
  explained in more detailed in the main text.

- Minimal sufficient explanations are not unique.  I strongly suggest the
  authors to discuss uniqueness of the computed explanations in the Limitations
  paragraph.

**Other Strengths And Weaknesses:**

#STRENGTHS

- Clearly written and structured
- The motivation is likewise clear
- Presents a sensible and technically non-trivial solution
- The experiments are well designed
- The results provide evidence that the claims hold

#WEAKNESSES

- Even when tackled using nice strategies -- like the proposed algorithm -- finding provably sufficient explanations is still computationally challenging, limiting (at least at present) applicability and significance.

**Questions For Authors:**

- Could your algorithm help *enumeration* of minimal sufficient explanations too?  If so, it may be worth to point this out, as a number of papers on formal explainability are concerned with this very problem.

**Relation To Broader Scientific Literature:**

To the best of my knowledge, the contribution is positioned appropriately against existing literature.

**Theoretical Claims:**

I did not check the correctness of the claims.  They all seem to be intuitively
reasonable.

---

> ### Author Rebuttal · Authors · 2025-04-01
>
> We appreciate the reviewer's constructive feedback and their acknowledgment of the significance of our work.
>
> **Enumeration of minimal sufficient explanations**
>
> We thank the reviewer for the insightful comment and agree that this is a highly relevant direction, particularly in the context of formal explainable AI. We touch on a related aspect in Appendix C, where we explore how varying feature orderings can influence the explanations generated by our algorithm. Intuitively, running the algorithm over different orderings yields different explanations. A more advanced strategy could start with an initial explanation and iteratively perturb specific features - e.g., replacing features in the explanation with those from its complement - to produce alternative explanations. Another possible direction is to explicitly maximize the divergence between explanations to encourage diversity, for instance by using reverse orderings. While enumerating all possible explanations is computationally intensive, one could potentially leverage the duality with the minimum hitting set (MHS) problem [1] to generate many contrastive explanations and iterate over diverse MHSs. However, such an approach may not scale well to large input domains due to the sheer number of possible contrastive explanations. We appreciate the importance of this point and will incorporate a discussion of it into both the main text and appendix of the final version, and highlight it as an interesting direction for future research.
>
> **Additional weaknesses, comments and suggestions**
>
> We first acknowledge the weakness pointed out by the reviewer regarding the scalability challenges in producing provably sufficient and minimal explanations for neural networks. However, as both the reviewer and other reviewers have noted, our method represents a significant improvement over prior approaches tackling the same task - offering better computation time, support for larger model sizes, and more concise explanations. We also refer the reviewer to Appendix D, where we demonstrate the generalizability of our method across additional settings. This includes evaluation on a language benchmark derived from a medical NLP dataset, where certification is conducted using meaning-preserving perturbations within an embedded input space, as well as in a regression setting. In the latter, we use the real-world, safety-critical Taxi-Net benchmark to show that fixing certain input features keeps predictions within a target range.
>
> Moreover, we highly appreciate the reviewer’s many thoughtful comments and suggestions, which we will incorporate into the final version and will help us improve the clarity of our work. Regarding Algorithm 2, the counterexample is obtained directly from the verifier but can also be extracted through an external adversarial attack, and we will clarify this accordingly. By a standard algorithm, we refer to a typical greedy approach, which does not incorporate abstraction-refinement, such as the one presented in Algorithm 1 and used in prior works (e.g., the method by Wu et al., NeurIPS 2023 [2]). Importantly, we emphasize that our abstraction-refinement approach is agnostic to the specific technique used to generate explanations and is applicable to any method that derives explanations through multiple certification queries. We agree that this point should be more clearly articulated in the final version.
>
> We agree with the reviewer’s comment on the importance of improving the discussion of the abstraction methodology in the main text. To enhance clarity, we will relocate portions of the detailed expedition of the abstraction technique from Appendix A to Section 3 and refine the overall discussion accordingly. Additionally, the standard deviation is computed as the square root of the mean of the squared deviations from the average, normalized by dividing by the total number of data points used in each separate experiment. We will also address the notation issues raised and emphasize the non-uniqueness of the generated subsets. While this aspect is partially covered in Appendix C, which discusses different feature orderings, we will ensure it is explicitly highlighted and discussed in the main text as well. Thank you for bringing these valuable points to our attention!
>
> Lastly, we appreciate the reviewer’s interesting point regarding the non-monotonicity that is observed in Table 1. This effect is due to the experiment's timeout: without it, the behavior remains monotonic (see Fig. 6). However, with the timeout, larger abstraction configurations can get stuck earlier in the algorithm, causing them to produce larger explanations within the time limit, which explains the observed slight non-monotonic behavior. We will clarify this interesting point in our final version.
>
> [1] From Contrastive to Abductive Explanations and Back Again (Ignatiev et al., KR 2021)
>
> [2] Verix: Towards Verified Explainability of Deep Neural Networks (Wu et al., Neurips 2023)

---

### Official Review · Reviewer_jftS · 2025-03-09

**Overall Recommendation:** 4

**Summary:**

This paper aims to improve the scalability of verification algorithms for computing minimal sufficient explanations of neural network predictions. \
The traditional approach iteratively removes input features while preserving the invariance property that the retained set must remain a sufficient explanation. However, this strategy is computationally expensive due to the large number of evaluations required for the neural network. \
The proposed approach introduces an abstraction-refinement technique that enables querying a surrogate, smaller network, thereby reducing the computational cost of verification. The approach progressively increases the network size to ensure convergence to the minimal sufficient explanation set. Notably, the ordering of network sizes corresponds to an ordering of the explanation sets (under the subset relation), allowing formal guarantees such as minimality and sufficiency. \
Experiments are conducted on three datasets, MNIST, CIFAR-10 and GTSRB, comparing the proposed method against the traditional formal approach and demonstrating improved efficiency. Additionally, comparisons with heuristic-based methods, including Anchors and SIS, highlight the ability to reduce the computational time gap between formal and heuristic-based approaches while ensuring formal guarantees.

## update after rebuttal

The authors have addressed my concerns. I think the additional effort put into the rebuttal have enhanced the completeness of the work, in particular with respect to the additional computational analysis of querying surrogate oracles, the analysis of the scalability trend (scaling the size of the original backbone) and the additional experiments to make the analysis consistent across different network architectures (in terms of activation functions). All these experiments should be included in the paper or in the supplementary material. Additionally, the illustrative examples are an interesting addition to improve the clarity of the presentation and increase the accessibility and readability of the paper. These examples should be included in the main paper.
Overall, I'm happy to increase my score. Congratulations to the authors !

**Claims And Evidence:**

Overall, the claims of the paper are clear and reasonable. The idea of leveraging a continuum of progressively larger models to reduce the computation time for verification and improve the scalability is novel and significant.

**Essential References Not Discussed:**

No

**Experimental Designs Or Analyses:**

Some of the design choices for the experimental analysis are questionable. For instance the backbone network for evaluation uses sigmoid activations for MNIST and GTSRB, and ReLU for CIFAR-10. In order to improve the completeness of the analysis, experiments with both sigmoid and ReLU should be provided in all datasets.
Additionally, it would be good to perform an analysis with networks using residual connections, otherwise the scope of validity of the results is quite limited.

In addition to Figure 4, it would be good to provide a more fine-grained analysis of the computation time, including the number of call evaluations to the neural network surrogates with the corresponding cost in terms of time per call (over iteration/feature).

Additionally, the analysis doesn’t provide any insight about the “trend” for the scalability, as the size of the original backbones is kept fixed. It would be good to repeat the analysis for smaller models (or larger backbones) and compare with the already provided results.

**Methods And Evaluation Criteria:**

Experiments are in my opinion overall convincing, but should be strengthened in terms of their scope. Please refer to the Experimental Designs Or Analyses Section for further details.

**Other Comments Or Suggestions:**

None

**Other Strengths And Weaknesses:**

The paper can be improved in terms of Clarity. Specifically:
1. It would be good to provide a running example (simple yet non-trivial) to highlight the notion of abstraction and refinement of a network and the corresponding consequence on the explanation set. A lot of definitions and propositions are provided without giving the reader the possibility to gain an intuition.
2. No sufficient detail about the proving of sufficiency and the way in which networks are abstracted and refined is provided. This hinders the clarity and reproducibility of the work.

Overall, the paper makes an interesting and original contribution. However, the scope of the experimental analysis is rather limited and the paper can improve in terms of clarity. Therefore, I make an initial and cautious judgement, but I'm willing to increase my score.

**Questions For Authors:**

Please refer to the Sections:
- Experimental Designs Or Analyses
- Other Strengths And Weaknesses

**Relation To Broader Scientific Literature:**

In my opinion, there is good discussion about the related work.

**Theoretical Claims:**

I haven’t checked the proofs for the theory, but all results about sufficiency and minimality are reasonable. There is however a major issue with the clarity of the presentation. Please refer to the Other Strength and Weaknesses section for further details.

---

> ### Author Rebuttal · Authors · 2025-04-01
>
> We appreciate the reviewer's constructive feedback and their acknowledgment of the significance of our work.
>
> **Expanding the scope**
>
> We thank the reviewer for bringing up this point and will improve its discussion in the final version. We note that while our primary experiments focus on vision classification models, our approach is domain-agnostic and can be applied to other domains. In Appendix D, we provide example experiments in additional domains. For language tasks, we utilized the only language-specific benchmark from the annual neural network verification competition (VNN-COMP [1]), which is based on a medical safety NLP dataset. Certification was achieved over an embedded representation of the input space, ensuring meaning-preserving perturbations. Another example of an expansion can include regression tasks, not just classification. In this case, the provable guarantee would ensure that fixing a subset of features keeps the output prediction within a specified range. We demonstrate this in Appendix D using the real-world, safety-critical Taxi-Net benchmark from Wu et al. (Neurips 2023, [2]). We will enhance our discussion of these extensions, as well as other potential applications, in the body of the final draft.
>
> **Design choices for experimental analysis**
>
> We selected these specific configurations because they are utilized in the annual neural network verification competition (VNN-COMP [1]). However, in response to the reviewer’s feedback, we will include the additional requested configurations in our final version, with the varying distinct activation functions used for each setting. Our method is also compatible with networks that include residual connections, and we will include an experiment in the final version to highlight this.
>
> **Performance trend and additional quantitative measures**
>
> We thank the reviewer for these valuable suggestions. We will indeed incorporate additional quantitative measures in the final version, specifically detailing both time per query and time per iteration across our various settings. Due to space limitations in the rebuttal, we provide here only representative results - namely, the average query times in seconds for our different settings in Section E.5.
>
> | Abstraction| MNIST | CIFAR | GTSRB |
> |-------|-------|-------|-------|
> | $\rho$=0.1 | 0.08 | 0.35 | 1.89  |
> | $\rho$=0.2 | 0.11 | 0.42 | 3.05  |
> | $\rho$=0.3 | 0.13 | 0.55 | 4.73  |
> | $\rho$=0.4 | 0.16 | 0.75 | 7.05  |
> | $\rho$=0.5 | 0.18 | 0.90 | 9.27  |
> | $\rho$=0.6 | 0.20 | 0.92 | 11.28 |
> | $\rho$=0.7 | 0.22 | 0.93 | 12.97 |
> | $\rho$=0.8 | 0.23 | 0.92 | 14.38 |
> | $\rho$=0.9 | 0.25 | 0.93 | 15.68 |
>
> We appreciate the reviewer’s suggestion to examine the scalability trend in our results and will include dedicated experiments in the final version to explore this aspect more thoroughly. Our current findings indicate that increasing model size leads to an increase in overall generation time. At the same time, this also amplifies the relative benefits of the abstraction-refinement strategy, as larger models tend to produce more significant improvements when using coarser abstractions. To illustrate this effect more clearly, we performed an additional experiment based on the “Marabou” benchmark from VNN-COMP [1]. While our original setup focused solely on the “cifar10_large” model, we now include the “cifar10_medium” and “cifar10_small” variants as well. In this experiment, the abstraction-refinement approach showed relative improvements over the standard method, reducing computation time by 136.46 seconds for the small model, 264.18 seconds for the medium model, and 570.39 seconds for the large model. In the final version, we will include additional experiments that vary model sizes while keeping all other parameters constant, to more clearly illustrate this trend.
>
> **Paper clarity improvements**
>
> We thank the reviewer for the helpful feedback on these points. In the final version, we will include a running example to help clarify our definitions. Due to space limitations, our current detailed explanation of the sufficiency proofs and abstraction appears primarily in Appendix A, with only a concise summary in the main text. As recommended, we will integrate portions of this discussion into the main body and improve the overall presentation of these topics.
>
> [1] First three years of the international verification of neural networks competition (VNN-COMP) (Brix et al., STTT 2023)
>
> [2] Verix: Towards Verified Explainability of Deep Neural Networks (Wu et al., Neurips 2023)

---

> > ### Comment · Reviewer_jftS · 2025-04-07
> >
> > Thank you for the answers. I appreciate the additional computational analysis of querying the surrogate oracles and the analysis of the scalability trend, which contribute to strengthen the approach. There are still some aspects that are not adequately addressed. Specifically, the experimental analysis on the network choices should be made consistent (both in terms of activation and architecture) in order to assess the scope of validity of the proposed solution. Moreover, an intuitive example would be appreciated to enhance the clarity of the paper.

---

> > > ### Author Response · Authors · 2025-04-09
> > >
> > > We thank the reviewer for their response. We are glad to hear that some points have been addressed and agree that they will help strengthen our work. We aim to clarify the remaining open points below:
> > >
> > > **Additional experimental analysis for consistency**
> > >
> > > We agree with the reviewer that including alternative configurations with different activation functions is important for consistency, even though our initial choices were based on architectures used in VNN-COMP. In response, we added the requested experiments with both ReLU and Sigmoid activations in Section 5.1. The results from the complementary variants are as follows: For MNIST with ReLU, our method achieved an average runtime of 46.88s vs. 52.68s for the standard method. With a 20s timeout, our explanations averaged 380.36 in size, compared to 439.74—showing improvements in both speed and explanation size. For CIFAR with Sigmoid, our method ran in 411.29s on average, versus 999.14s for the standard method, and yielded significantly smaller explanations (309.13 vs. 576.50). On GTSRB with ReLU, due to the time constraints of the rebuttal phase, we were only able to run a partial experiment (with a 2000s timeout). This yielded explanation sizes of 701.0 (ours) compared to 849.4 (standard). Despite currently being partial, these results too already point to a significant improvement in explanation size and an expected runtime improvement of the full experiment.
> > >
> > > In the final version, we will include the full experiment with a thorough analysis of all results, presenting them in both detailed tables and through the relevant visualizations and ablations—similar to those provided for the other benchmarks. We appreciate the reviewer for raising this important point.
> > >
> > >
> > > **Incorporating running examples**
> > >
> > > We thank the reviewer for this valuable suggestion. We will add a running example to the main text and move parts of Appendix A into Sections 3 and 4. We also share a simple [illustrative example](https://postimg.cc/yDjjKCfJ), which we will expand upon in the final version. It demonstrates several concepts using a toy ReLU network with positive weights and biases, allowing for exact bounds via simplified interval-bound propagation. While intentionally simplified, the example serves to illustrate the procedure.
> > >
> > > Figure (a) shows the original model and the interpreted input (0,1,1). All biases are set to 0 except for the lower output neuron, which has a bias of 10. Propagating (0,1,1) gives outputs of 15 (class 1) and 46 (class 2), so class 2 is predicted. Figure (b.1) illustrates an explanation that includes features 2 and 3 (in orange): we fix these features to their original values of 1, restricting their domains to [1,1]. Feature 1 is not in the explanation and hence is allowed to vary freely in the range [0,1]. We then compute bounds via interval propagation. For example, the top hidden neuron gets an input range of [3, 5], from a lower bound of 0x2 + 1x2 + 1×1 = 3 and an upper bound of 1x2 + 1x2 + 1×1 = 5. These bounds are propagated to the output layer using the weights. For example, the top output neuron's range is [15, 22], calculated as: lower bound = 3x2 + 3×1 + 6×1 = 15 and upper bound = 5x2 + 5×1 + 7×1 = 22. Overall, the output range for class 2 ([46,55]) is strictly above that of class 1 ([15,22]). Therefore, fixing features 2 and 3 is *sufficient* to guarantee that class 2 remains the predicted class—making it a valid explanation.
> > >
> > > In Figure (b.2), we illustrate how three hidden neurons are merged by unifying their intervals and computing a weighted sum. While in practice we use the Minkowski sum to obtain a tighter bound, we simplify the process here for clarity. For example, the top neuron has bounds of 3 × (2+1+1) = 12 and 7 × (2+1+1) = 28, giving the interval [12, 28]. Since this lies strictly below [31, 59], features 2 and 3 form an “abstract explanation” per Definition 3. Moreover, figures (c.1) and (c.2) show that feature 3 alone is also an explanation. While it's a *minimal* explanation for the original model, it isn't one for the abstract model, since [4, 28] and [17, 59] overlap—violating Definition 3. This shows that while every abstract explanation is valid for the original model, minimal explanations may not be.
> > >
> > > Figures (d.1) and (d.2) show a refinement where only the first two neurons are merged, resulting in two merged neurons: one for the first and second, and one for the third. The output interval [37, 55] remains strictly above [8, 22], confirming that fixing only feature 3 is a *minimal* explanation for both the *refined and original model*.
> > >
> > > We emphasize that the most significant improvements in our method appear in much more complex models with tighter bounds and harder certification. The examples shown were deliberately simple to clarify the general methodology, which we agree is important to illustrate intuitively.
> > >
> > > Once again, we thank the reviewer for these very important remarks and for helping us improve our work!

---

### Official Review · Reviewer_pUMo · 2025-03-13

**Overall Recommendation:** 3

**Summary:**

This paper introduces a novel abstraction-refinement technique to efficiently compute provably sufficient explanations of neural network predictions, defined as a subset of input features that are sufficient to determine that the prediction remains the same. The method constructs an abstract neural network, which is significantly smaller than the original model, by merging similar neurons, showing that a sufficient explanation for the abstract model is also provably sufficient for the original. Since an explanation that is minimal for the abstract network may not be minimal for the original, authors introduce an approach that iteratively refines the abstract network by gradually increasing its size until a provably minimal sufficient explanation is found for the refined network that is also provably minimal for the original. This method substantially improves the efficiency of computing sufficient explanations compared to the existing verification-based baseline while also outperforming heuristic-based methods –in terms of explanation size and time-- which fail to provide sufficient explanations and lack formal guarantees. The results demonstrate that the approach enhances scalability, interpretability, and flexibility, offering a more fine-grained understanding of neural network decisions.

**Claims And Evidence:**

The main claim that the proposed method outperforms existing verification-based methods by producing smaller explanations more efficiently is supported by the evaluation setup. Also, the comparison against heuristic-based approaches shows that they do not provide sufficient explanations while lacking theoretical guarantees.
However, I still find the following claims unsupported:
1.	Claim: “A smaller explanation provides a better interpretation, and for this reason, the minimality of the explanation is also a desired property” in L019-023.
Issue: this claim is not well-supported by the authors throughout the paper. Since this is the basis behind finding provably minimal sufficient explanations (the main contribution in the paper), I think it would be more convincing to add how this improves interpretability (e.g., visually, quantitatively) and compare it to other non-minimal explanations.

**Essential References Not Discussed:**

Not that I am aware of.

**Experimental Designs Or Analyses:**

Yes. I checked the experimental design described in Section 5 (Experimental results) and Appendix B, C and D.
One issue is the choice of the perturbation radius $\epsilon_p$. There is no justification as to why the values were chosen to be 0.01 and 0.001 (L734-L735).

**Methods And Evaluation Criteria:**

The proposed method for providing provably sufficient explanations efficiently is well-motivated, especially for safety-critical domains, where having reliable explanations is crucial.

However, there is quite a few limitations in the evaluation criteria:
-	Limited diversity in benchmark datasets and architecture: the evaluation focuses on MNIST, CIFAR-10 and GTSRB, but lacks a broader range of complex real-world datasets that are more relevant for downstream tasks.
-	Lack of qualitative comparison of explanations: the paper only evaluates their method against heuristic-based methods quantitatively in Table 2. (e.g., size and efficiency), but does not provide a qualitative comparison. It is unclear whether the generated provably sufficient and minimal explanations are more meaningful semantically or more interpretable than heuristic-based explanations.
-	Quantitative evaluation metrics are not enough: The metrics used (e.g. explanation size and time) are not enough from an explainability point of view. Perhaps it would be useful to include quantitative interpretability metrics (e.g., Grid Pointing Game), or any other similar metric, to see how provably sufficient explanations can truly localize class-discriminative features compared to heuristic-based approaches.

**Other Comments Or Suggestions:**

no further suggestions

**Other Strengths And Weaknesses:**

Strengths
-	This paper introduces a novel abstraction-refinement approach to efficiently generate provably sufficient explanations for neural networks. The idea is interesting and reasonable: to show that a certain property of a reduced size model (e.g., abstract sufficient explanation) provably holds for the original model, especially for the verification literature which is often limited by scalability
-	The paper is well-organized, and the description of the algorithm and evaluation is clear.

Weaknesses
-	Certified radius: there is no evaluation on the effect of changing the perturbation radius on the produced explanations. Clearly a larger radius would be more desired, but there is no elaboration on this part.
-	Halting condition: there are no clear evaluation criteria or recommendation of which halting condition (e.g., network size $\rho$) a user should decide on to improve both the computation time and interpretability. (L090-092, right)

**Questions For Authors:**

1.	How does the proposed method perform w.r.t different perturbation radii $\epsilon_p$? I think this is a crucial point to show, as to also justify the choice of the current radii.

**Relation To Broader Scientific Literature:**

This paper contributes to the field of formal explainable artificial intelligence (formal XAI) by leveraging formal verification techniques (abstraction refinement) to provide provably sufficient explanations more efficiently. Unlike heuristic-based methods such as Anchors and SIS, which lack guarantees, and prior verification techniques that are computationally expensive, this method merges neurons to reduce verification complexity while ensuring explanation sufficiency. The paper extends work in neural network verification (e.g., Reluplex, SMT solvers) by refining the abstraction iteratively to find a minimal sufficient explanation. This improves over the verification baseline method (Alg. 1), making formal guarantees less computationally expensive with applications in trustworthy AI and safety-critical domains. By bridging verification, explainability, and scalability, this work contributes to making formal XAI more practical.

**Theoretical Claims:**

Claim: Alg. 2 is claimed to produce a provably sufficient and minimal explanation as stated in Proposition 5.

Issue:  Going through the proof, I did not find any detail on whether the final explanation is always the globally minimal one, or whether there are multiple minimal explanations (different subsets of input features of same minimal size). If so, this needs to be explicitly clarified.

---

> ### Author Rebuttal · Authors · 2025-04-01
>
> We appreciate the reviewer's constructive feedback and their acknowledgment of the significance of our work.
>
> **Extension to different $\epsilon_p$ perturbations**
>
> We thank the reviewer for raising this important point. As correctly noted, larger $ϵ_p$ perturbations yield stronger sufficiency guarantees but may result in larger explanations. We chose the perturbation levels in this paper based on those commonly used in the annual Neural Network Verification Competition (VNN-COMP [1]), but our method is general and can be applied to any perturbation. We already include an ablation study in *Appendix C*, which we’ll reference more clearly. Per the reviewer’s suggestion, we’ll expand it with additional benchmarks, perturbation levels, and visualizations.
>
> **Qualitative analysis and the importance of minimality**
>
> We agree that this is an important point, and will include further qualitative assessments in the final version. As a preliminary example, we share one [illustrative case](https://postimg.cc/Mcg8tp1y).
>
> First, image (a) highlights the importance of minimality. For example, the interior of the forward sign (red frame in image (c)) is not included in our explanation, and the edges alone are sufficient for classification. This shows *the interior pixels are irrelevant* and can be excluded without affecting the prediction. In contrast, non-minimal sufficient subsets (e.g., the other examples) include unnecessary features. To better grasp the significance of small or minimal explanations, one can consider the extreme case where the entire image is chosen as the explanation: although it is clearly sufficient, it is neither minimal nor informative.
>
> Second, image (b) shows that fixing the provably sufficient subset ensures robustness: any change in the complement $\bar{S}$ within the domain doesn't alter the classification. This contrasts with heuristic subsets, where changes in the complement can flip the prediction - revealing their limitations. The explanation also matches human intuition, as the highlighted region (the triangular forward sign) alone justifies the predicted class, unlike the much less clear heuristic explanations.
>
> **Additional quantitative analysis and scope**
>
> Since our work focuses on *sufficient explanations* rather than the more widely studied *additive attributions*, certain commonly used evaluation metrics - such as infidelity [2], which are specifically designed for additive forms - are not directly applicable. Adapting such metrics for sufficiency-based explanations would require significant modifications and, on its own, constitutes an interesting direction for future work. Therefore, we followed established conventions in the literature [e.g., 3–5] for evaluating sufficiency-based explanations, relying on the three widely accepted metrics in this area: generation time, sufficiency, and conciseness.
>
> That said, we appreciate the reviewer’s interesting idea of assessing how well sufficient explanations localize class-discriminative features. We tested this using object-detection ground truth from GTSRB and found that indeed 93.33% of our method’s explanation pixels aligned with annotated regions, outperforming Anchors (60.36%). We thank the reviewer for this idea and will include a detailed experiment on this point in the final version.
>
> Lastly, while our main experiments focus on vision classification models, our approach is domain-agnostic and can extend to other domains. Appendix D includes examples of both language and regression tasks. For language, we use the only VNN-COMP [1] language benchmark based on a medical NLP dataset, certifying meaning-preserving perturbations via an embedded input space. For regression, we use the real-world safety-critical Taxi-Net benchmark used in Wu et al. (Neurips 2023, [3]), showing that fixing input features keeps predictions within a target range. We will expand on these extensions and additional applications in the final draft.
>
> [1] First Three Years of the International Verification of Neural Networks Competition (VNN-COMP) (Brix et al., STTT 2023)
>
> [2] On the (In) Fidelity and Sensitivity of Explanations (Yeh et al., Neurips 2019)
>
> [3] Verix: Towards Verified Explainability of Deep Neural Networks (Wu et al., Neurips 2023)
>
> [4] On Guaranteed Optimal Robust Explanations for NLP Models (La Malfa et al., IJCAI 2021)
>
> [5] Abduction-based Explanations for Machine Learning Models (Ignatiev et al., AAAI 2019)

---

### Official Review · Reviewer_ztpX · 2025-03-20

**Overall Recommendation:** 3

**Summary:**

The paper seeks to use an abstraction-refinement approach to generate "provably sufficient" explanations for neural networks.

==
The rebuttal has satisfactorily addressed my major concerns.

**Claims And Evidence:**

1. The paper is motivated by the need for proofs in high-assurance systems. However, the investigations are on relatively simple benchmarks and simple models.

2. The notion of a provably correct AI system is clear. However, provably sufficient explanations need not carry the same sense of high assurance.

3. Explanations are often meant for human users. It is not clear that a provably sufficient explanation will be desired by a human end user or will enhance assurance arguments for complex systems.

**Essential References Not Discussed:**

None

**Experimental Designs Or Analyses:**

1. The experiments are on really small models and low-resolution data sets.

**Methods And Evaluation Criteria:**

1. The method is evaluated on relatively simple benchmarks using simple models.
2. There is no evaluation to show these provably sufficient explanations lead to high assurance, which is the central motivation of the paper.

**Other Comments Or Suggestions:**

None

**Other Strengths And Weaknesses:**

None

**Questions For Authors:**

None

**Relation To Broader Scientific Literature:**

The paper introduces ideas from formal verification such as abstraction refinement to explainable AI.

**Theoretical Claims:**

1. The theoretical claims primarily build upon earlier work on abstraction refinement and neural network verification.
2. The connection between explainable AI for high-assurance applications and provably sufficient explanations is tenuous.

---

> ### Author Rebuttal · Authors · 2025-04-01
>
> We thank the reviewer for their valuable and constructive feedback.
>
> **Improving the motivation behind the work**
>
> We agree that the motivation for obtaining explanations with provable guarantees, as well as the importance of the specific guarantees we provide, can be better articulated. We will make sure to better clarify this in the final version. Please see our detailed response on this point below:
>
> **General motivation.** Explanations for black-box models are often sought to *enhance user trust* in the model. For instance, consider a scenario where a medical professional must trust a black-box model to diagnose a disease from MRI scans. An explanation can help them decide whether to rely on the prediction - especially if it highlights medically relevant image regions. However, if the explanation is untrustworthy and lacks guarantees - such as presenting a subset that isn’t truly sufficient - it can misrepresent the model’s reasoning, leading the medical professional to place misplaced trust in it, potentially resulting in harmful outcomes.
>
> Such scenarios have sparked increasing interest in the literature in developing explanations that come with provable guarantees [see, e.g., 1–8]. In this context, *explanations that are both provably minimal and sufficient* have gained significant attention as a highly sought-after form of provable explanation [see, e.g., 1-8, among others]. The key idea is to identify small subsets of features that, even when all other features take arbitrary values, still preserve the model’s prediction. This ensures the subset’s certified sufficiency, helping users understand the reasoning behind the prediction and disregard irrelevant features.
>
> **Scope.** While we agree that the guarantees our method provides inherently face scalability challenges - due to their reliance on verification, as is with all approaches tackling this task - we emphasize the significant relative improvements our approach achieves over previous methods. As several reviewers have also positively noted, our contributions are particularly notable in terms of reduced computation time, the generation of smaller explanations, and the ability to handle larger models. As the field of neural network verification continues to progress rapidly [9-11], our method - offering a significant, orthogonal enhancement for generating explanations using these tools - will similarly evolve.
>
> Moreover, to further show the generality of our results, we present experiments on two additional benchmarks in Appendix D: (1) the only language task from the annual neural network verification competition (VNN-COMP, [11]), based on a medical safety NLP dataset, which uses meaning-preserving perturbations in an embedded input space; and (2) a regression task, where our guarantees ensure output stability, evaluated on the *real-world, safety-critical Taxi-Net benchmark*, evaluated in Wu et al. (Neurips 2023, [2]).
>
> Finally, in response to the reviewer’s comment and reviewer pUMo’s feedback, we will include additional qualitative examples in the final version to better motivate our work. As part of our response to reviewer pUMo, we have added a small illustrative visualization, which we also plan to incorporate - along with other visualizations - into the final version.
>
> We thank the reviewer for highlighting this important point and will significantly enhance our discussion of these aspects in the final version.
>
> [1] Delivering Trustworthy AI Through Formal XAI (Marques Silva et al., AAAI 2022)
>
> [2] Verix: Towards Verified Explainability of Deep Neural Networks (Wu et al., Neurips 2023)
>
> [3] Abduction-Based Explanations for Machine Learning Models (Ignatiev et al., AAAI 2019)
>
> [4] On Guaranteed Optimal Robust Explanations for NLP Models (La Malfa et al., IJCAI 2021)
>
> [5] Model Interpretability Through the Lens of Computational Complexity (Barcelo et al., Neurips 2020)
>
> [6] Computing Abductive Explanations for Boosted Trees (Audemard et al., AISTATS 2023)
>
> [7] Explanations for Monotonic Classifiers (Marques Silva et al., ICML 2020)
>
> [8] Foundations of Symbolic Languages for Model Interpretability (Arenas et al., Neurips 2021)
>
> [9] Beta-Crown: Efficient Bound Propagation with Per-Neuron Split Constraints for Neural Network Robustness Verification (Wang et al., Neurips 2021)
>
> [10] Scalable Neural Network Verification with Branch-and-bound Inferred Cutting Planes (Zhou et al., Neurips 2024)
>
> [11] First Three Years of the International Verification of Neural Networks Competition (VNN-COMP) (Brix et al., STTT 2023)

---

### Decision · Program_Chairs · 2025-05-01

**Decision:**

Accept (poster)

**Comment:**

In the context of post-hoc explainability, this paper addresses the challenge of computing minimal sufficient reasons for the classifications made by neural networks. Here, sufficiency is assessed based on the neighborhood of the queried instance. Since solving this problem is generally NP-hard, the authors propose an abstraction-based explanation approach that starts with a coarse approximation of the neural net and iteratively refines this abstraction until a minimal sufficient reason is identified. The authors demonstrate the correctness of their approach through various results and validate its practical performance, measured in terms of runtime and explanation sizes, across several benchmarks.

The average score for this paper exceeds the acceptance threshold, with reviewers providing scores that range from "weak accept" to "accept." All reviewers acknowledge the novelty of this abstraction approach, noting the intriguing connections it establishes between formal explainability and model abstraction. Furthermore, revised reviews indicate that the authors effectively addressed concerns regarding the scope of the study, the experimental design choices, and the need for additional statistical tests. For these reasons, I recommend accepting the paper.

In a revised version, I suggest incorporating constructive feedback from the rebuttal phase to further enhance the paper and address the concerns raised. Additionally, adding comments about the runtime complexity of the neuron-merging algorithm in Section 3 would provide valuable insights into the feasibility of this approach.